# Lifelong exposure to high-altitude hypoxia in humans is associated with improved redox homeostasis and structural–functional adaptations of the neurovascular unit

Benjamin S. Stacey[1] , Ryan L. Hoiland[2,3] , Hannah G. Caldwell[4] , Connor A. Howe[4] ,
Tyler Vermeulen[4] , Michael M. Tymko[4,5,6] , Gustavo A. Vizcardo-Galindo[7] , Daniella Bermudez[7] ,
Rómulo J. Figueroa-Mujíica[7] , Christopher Gasho[8], Edouard Tuaillon[9], Christophe Hirtz[10] ,
Sylvain Lehmann[10], Nicola Marchi[11], Hayato Tsukamoto[12] , Francisco C. Villafuerte[7] ,
Philip N. Ainslie[1,4] and Damian M. Bailey[1]

[1]*Neurovascular Research Laboratory, Faculty of Life Sciences and Education, University of South Wales, Pontypridd, UK*

[2]*Department of Anaesthesiology, Pharmacology and Therapeutics, Vancouver General Hospital, University of British Columbia, Vancouver, British Columbia, Canada*

[3]*Department of Cellular and Physiological Sciences, Faculty of Medicine, University of British Columbia, Vancouver, British Columbia, Canada*

[4]*Centre for Heart, Lung and Vascular Health, University of British Columbia-Okanagan Campus, Kelowna, British Columbia, Canada*

[5]*Faculty of Kinesiology, Sport, and Recreation, University of Alberta, Edmonton, Alberta, Canada*

[6]*Department of Medicine, Faculty of Medicine, University of British Columbia, Vancouver, British Columbia, Canada*

[7]*Laboratorio de Fisiología Comparada, Departamento de Ciencias Biológicas y Fisiológicas, Facultad de Ciencias y Filosofía, Universidad Peruana Cayetano Heredia, Lima 31, Peru*

[8]*Division of Pulmonary and Critical Care, Loma Linda University School of Medicine, Loma Linda, CA, USA*

[9]*Department of Infectious Diseases, University of Montpellier, Montpellier, France*

[10]*LBPC-PPC, Université de Montpellier, IRMB CHU de Montpellier, INM INSERM, Montpellier, France*

[11]*Laboratory of Cerebrovascular and Glia Research, Department of Neuroscience, Institute of Functional Genomics, University of Montpellier, Montpellier, France*

[12]*Faculty of Sport and Health Science, Ritsumeikan University, Kusatsu, Shiga, Japan*

Handling Editors: Laura Bennet & Ken O'Halloran

The peer review history is available in the Supporting Information section of this article (https://doi.org/10.1113/JP283362#support-information-section).

**Benjamin S. Stacey** is a Lecturer in Clinical Science at the University of South Wales and member of the Neurovascular Research Laboratory led by Professor Damian Bailey where he completed his PhD. **Damian M. Bailey** is a Royal Society Wolfson Research Fellow and Professor of Physiology and Biochemistry at the University of South Wales. He is Editor-in-Chief of *Experimental Physiology* and Chair of the Life Sciences Working Group to the European Space Agency. The Neurovascular Research Laboratory takes an integrated translational approach to investigate how free radicals and associated reactive oxygen/nitrogen species control substrate delivery to the brain across the clinical spectrum of human health and disease.

B. S. Stacey and D. M. Bailey contributed equally to this work.

The Journal of Physiology

**Abstract** High-altitude (HA) hypoxia may alter the structural–functional integrity of the neuro-vascular unit (NVU). Herein, we compared male lowlanders ($n = 9$) at sea level (SL) and after 14 days acclimatization to 4300 m (chronic HA) in Cerro de Pasco (CdP), Péru (HA), against sex-, age- and body mass index-matched healthy highlanders ($n = 9$) native to CdP (lifelong HA). Venous blood was assayed for serum proteins reflecting NVU integrity, in addition to free radicals and nitric oxide (NO). Regional cerebral blood flow (CBF) was examined in conjunction with cerebral substrate delivery, dynamic cerebral autoregulation (dCA), cerebrovascular reactivity to carbon dioxide ($CVR_{CO2}$) and neurovascular coupling (NVC). Psychomotor tests were employed to examine cognitive function. Compared to lowlanders at SL, highlanders exhibited elevated basal plasma and red blood cell NO bioavailability, improved anterior and posterior dCA, elevated anterior $CVR_{CO2}$ and preserved cerebral substrate delivery, NVC and cognition. In highlanders, S100B, neurofilament light-chain (NF-L) and T-tau were consistently lower and cognition comparable to lowlanders following chronic-HA. These findings highlight novel integrated adaptations towards regulation of the NVU in highlanders that may represent a neuroprotective phenotype underpinning successful adaptation to the lifelong stress of HA hypoxia.

(Received 1 June 2022; accepted after revision 20 December 2022; first published online 11 January 2023)
**Corresponding author** D. M. Bailey: Neurovascular Research Laboratory, Faculty of Life Sciences and Education, University of South Wales, Pontypridd, UK. Email: damian.bailey@southwales.ac.uk

**Abstract figure legend** How the hypoxia of high altitude across the temporal continuum of chronic through to life-long exposure impacts the neurovascular unit (NVU) phenotype and to what extent this is subject to altered redox homeostasis were determined. Basal multimodal biomarkers reflecting NVU structure and function were determined in lowlanders at sea level (lifelong normoxia) and after 2 weeks acclimatization to 4300 m (chronic hypoxia) in Cerro de Pasco, Péru (CdP) and compared to age/sex-matched highlanders native to CdP (lifelong hypoxia). Compared

to lowlanders at sea level, highlanders were characterized by elevated systemic nitric oxide (NO) bioavailability, improved dynamic cerebral autoregulation (dCA) and cerebrovascular reactivity to carbon dioxide ($CVR_{CO_2}$), and lower concentrations of S100B, ubiquitin carboxy-terminal hydrolase-L1 (UCH-L1) and total tau (T-tau) in the face of preserved cerebral bioenergetics, NVC and cognition. Collectively, these novel findings tentatively suggest that these integrated adaptations may collectively reflect a neuroprotective phenotype to cope with the lifelong stress of high-altitude hypoxia. ICA, internal carotid artery; MCA, middle cerebral artery; PCA, posterior cerebral artery; VA, vertebral artery.

**Key points**

- High-altitude (HA) hypoxia has the potential to alter the structural–functional integrity of the neurovascular unit (NVU) in humans.
- For the first time, we examined to what extent chronic and lifelong hypoxia impacts multimodal biomarkers reflecting NVU structure and function in lowlanders and native Andean highlanders.
- Despite lowlanders presenting with a reduction in systemic oxidative–nitrosative stress and maintained cerebral bioenergetics and cerebrovascular function during chronic hypoxia, there was evidence for increased axonal injury and cognitive impairment.
- Compared to lowlanders at sea level, highlanders exhibited elevated vascular NO bioavailability, improved dynamic regulatory capacity and cerebrovascular reactivity, comparable cerebral substrate delivery and neurovascular coupling, and maintained cognition. Unlike lowlanders following chronic HA, highlanders presented with lower concentrations of S100B, neurofilament light chain and total tau.
- These findings highlight novel integrated adaptations towards the regulation of the NVU in highlanders that may represent a neuroprotective phenotype underpinning successful adaptation to the lifelong stress of HA hypoxia.

## Introduction

The neurovascular unit (NVU) forms the functionally integrated cellular network responsible for maintaining the structural integrity of the blood–brain barrier (BBB) and regulation of cerebral blood flow (CBF) via neuro-vascular coupling (NVC) (Kaplan et al., 2020). Free radicals and associated reactive oxygen and nitrogen species have recently been identified as key regulators that contribute towards the preservation of neuronal, glial and vascular homeostasis (Bailey et al., 2020; Cobley et al., 2018). Maintained redox homeostasis and a structurally and functionally intact NVU are likely integral to successful high-altitude (HA) adapatation given the importance of coupling cerebral substrate (oxygen and glucose) supply against demand and establishing efficient clearance of carbon dioxide ($CO_2$) to which the human brain has evolved heightened sensitivity (Bailey, 2019b).

Impaired NVU structure and function encompasses reduced NVC, dynamic cerebral autoregulation (dCA) and cerebrovascular reactivity to $CO_2$ ($CVR_{CO_2}$), and increased BBB permeability. These integrated components have been shown to precede, or at least associate with, the disordered redox homeostasis underlying cognitive decline and dementia (Kisler et al., 2017; Nation et al., 2019; Sweeney et al., 2018; van de Haar et al., 2016; Zhou et al., 2019) and may collectively interfere with the

dynamics of HA acclimatization and predispose to HA illness to which the lowlander is especially prone (Bailey, Bartsch et al., 2009).

Basal CBF, substrate delivery and NVC are well maintained in lowlanders born and bred at sea level (SL) when exposed to HA hypoxia (Caldwell et al., 2017; Hoiland, Howe et al., 2019; Willie et al., 2015). However, emerging evidence suggests that BBB integrity may become compromised due to impaired cerebral autoregulation subsequent to local elevations in oxidative–nitrosative stress (OXNOS) reflected by a free radical-mediated reduction in vascular nitric oxide (NO) bioavailability (Bailey, Taudorf et al., 2009; Bailey, Bartsch et al., 2009). Furthermore, magnetic resonance imaging (MRI) evidence of extracellular (vasogenic) oedematous brain swelling (Kallenberg et al., 2007) combined with hemosiderin deposits (Kallenberg et al., 2008) implying erythrocyte extravasation has been collectively inter-preted to reflect BBB disruption (Bailey, Bartsch et al., 2009) predisposing to impaired cerebral bioenergetic function and cognitive decline (Bailey et al., 2019).

In contrast, populations indigenous to HA, such as Andean, Tibetan and Ethiopian highlanders, have been exposed for millennia to the opportunity for natural selection to 'fine-tune' redox regulation of cerebral bioenergetics allowing survival and reproduction in such harsh environments (Bailey et al., 2019;

Beall, 2007). However, to what extent severe life-long hypoxia impacts the NVU and the integrated structural–functional adaptations that serve to maintain cerebral bioenergetics and cognitive function has not previously been investigated. This is surprising given that an estimated ∼500 million people permanently reside above 1500 m (Tremblay & Ainslie, 2021).

Therefore, the primary aim of the present study was to determine how the hypoxia of HA across the temporal continuum of chronic through to lifelong exposure impacts the NVU phenotype and to what extent this is subject to altered redox homeostasis. Basal multimodal metrics of NVU structure and function encompassing molecular (OXNOS, BBB integrity), haemodynamic (CBF, NVC, dCA, $CVR_{CO_2}$) and clinical (cognition) biomarkers were determined in lowlanders at SL (lifelong normoxia) and after 2 weeks acclimatization to 4300 m (chronic hypoxia) in Cerro de Pasco, Péru (CdP) and compared to age- and sex-matched highlanders native to CdP (lifelong hypoxia). We hypothesized that compared to acclimatized lowlanders, highlanders would be characterized by superior molecular, haemodynamic and clinical function collectively reflected by (1) lower circulating concentrations of OXNOS and NVU proteins, the latter reflecting a structurally tighter and more intact BBB in the absence of astrocytic–neuronal injury, (2) elevated CBF, NVC, dCA and $CVR_{CO_2}$ and (3) preserved cognition.

## Methods

### Ethical approval

Experimental procedures were approved by the University of British Columbia (H17-02687, H18-01404) and Universidad Peruana Cayetano Heredia (101686) ethics committees. Participants provided written informed consent and all procedures conformed to the *Declaration of Helsinki*, except for registration in a database.

### Participants

All participants were subject to a comprehensive medical examination that included a thorough assessment of medical history. Exclusion criteria included those with overt cardiopulmonary disease (hypertension, diabetes, hypertriglyceridaemia, chronic obstructive pulmonary disorder), significant developmental delay or learning difficulties, diagnosis of any central neurological disease (aneurysm, stroke, transient ischaemic attack, epilepsy, multiple sclerosis) and psychiatric disorders including any history of traumatic brain injury. For highlanders, inclusion criteria specified that they were born and had

lived permanently at their resident altitude. None of the participants were prescribed medication or taking nutritional supplements including over-the-counter anti-oxidant or anti-inflammatory medications over the course of the study. Additionally, all lowlanders abstained from prophylactic medication including carbonic anhydrase inhibitors at HA, given their potential to alter redox metabolism (Bailey et al., 2012).

**Lowlanders.** Nine male lowlanders of European descent who had been born and lived below 1500 m were recruited into the study (Table 1). Lowlanders were examined at SL (lifelong SL: ∼344 m) and after 2 weeks of HA acclimatization (chronic HA; ∼4300 m above SL). All lowlander participants avoided living at HA (>1500 m) for at least 6 months prior to HA testing.

**Highlanders.** We also recruited nine age- and sex-matched healthy Andean (Aymara) participants (Table 1) who were born and have been permanently living in Cerro de Pasco (lifelong HA: ∼4300 m above SL) or the Pasco region and did not work in the mining industry or present with excessive erythrocytosis (haemoglobin (Hb) > 20 g·dL$^{-1}$) (Leon-Velarde et al., 2005). All documents were translated into Spanish and participants spoke with a Peruvian research assistant to facilitate any discussion prior to obtaining written informed consent in Spanish.

### Experimental design

All participants were instructed to arrive at the laboratory a minimum of 6 h post-prandial, 12 h post-caffeine/alcohol, and 24 h post-exercise (Thijssen et al., 2011) for all visits. Lowlanders were tested at SL (∼344 m) and following 2 weeks of HA residence after an initial rapid ascent (270 km; 7–8 h drive) in a cargo van (Tymko et al., 2020).

### Metabolic function

Participants were asked to refrain from physical activity, caffeine and alcohol and to follow a low nitrate/nitrite ($NO_3^-/NO_2^-$) diet 24 h prior to formal experimentation (Woodside et al., 2014) and were 12 h overnight fasted when they attended the laboratory. Blood samples were obtained without stasis following 30 min of supine rest to control for plasma volume shifts.

**Blood sampling.** Venous blood samples (∼25 mL whole blood) were obtained without stasis from an indwelling cannula located in a forearm antecubital vein into EDTA, lithium heparin-coated and serum-separator tube Vacutainers (BD Biosciences, Oxford, UK) before

**Table 1. Demographics**

| Group: | Lowlanders (n = 9) | | Highlanders (n = 9) |
|---|---|---|---|
| Exposure-Location: | Lifelong-SL | Chronic-HA | Lifelong-HA |
| **Anthropometrics** | | | |
| Age (years) | 28 ± 8 | | 27 ± 7 |
| Mass (kg) | 74.8 ± 7.6 | | 70.8 ± 14.4 |
| Stature (m) | 1.75 ± 0.06 | | 1.65 ± 0.08 |
| | | | *P = 0.006, d = −1.480 |
| | | | (*P = 0.011, r = 0.595) |
| BMI (kg·m$^{-2}$) | 24.3 ± 1.7 | | 26.1 ± 6.0 |
| **Haematology** | | | |
| Hb (g·L$^{-1}$) | 14.8 ± 0.8 | 16.9 ± 0.8 | 18.1 ± 2.1 |
| | | *P = <0.001, d = 3.072 | *P = <0.001, d = 2.086 |
| | | (*P = 0.008, r = 0.628) | (*P = 0.003, r = 0.677) |
| Hct (%) | 45 ± 3 | 52 ± 3 | 55 ± 6 |
| | | *P = <0.001, d = 3.075 | *P = <0.001, d = 2.267 |
| | | (*P = 0.008, r = 0.628) | (*P = 0.003, r = 0.676) |
| Viscosity (cP) | 3.99 ± 0.23 | 4.72 ± 0.48 | 5.74 ± 1.02 |
| | | *P = <0.001, d = 1.975 | *P = <0.001, d = 2.376 |
| | | (*P = 0.008, r = 0.628) | (*P = <0.001, r = 0.822) |
| | | | †P = 0.016, d = 1.267 |
| | | | (†P = 0.031, r = 0.500) |
| **Blood gases** | | | |
| PvO$_2$ (mmHg) | 37 ± 8 | 26 ± 8 | 34 ± 8 |
| | | *P = 0.007, d = −1.214 | |
| | | (*P = 0.015, r = −0.573) | |
| PvCO$_2$ (mmHg) | 50 ± 5 | 38 ± 3 | 41 ± 6 |
| | | *P = <0.001, d = −2.283 | *P = <0.001, d = −2.283 |
| | | (*P = 0.008, r = −0.628) | (*P = 0.008, r = −0.616) |
| pH (units) | 7.36 ± 0.03 | 7.42 ± 0.02 | 7.36 ± 0.03 |
| | | *P = <0.001, d = 2.021 | †P = <0.001, d = −2.486 |
| | | (*P = 0.008, r = 0.628) | (†P = <0.001, r = −0.803) |
| HCO$_3^-$ (mmol·L$^{-1}$) | 28.3 ± 1.5 | 24.4 ± 1.7 | 23.8 ± 2.6 |
| | | *P = 0.001, d = −1.802 | *P = <0.001, d = −2.134 |
| | | (*P = 0.008, r = −0.629) | (*P = 0.001, r = −0.721) |
| SpO$_2$ (%) | 98 ± 1 | 85 ± 3 | 84 ± 2 |
| | | *P = <0.001, d = −4.117 | *P = <0.001, d = −9.350 |
| | | (*P = 0.008, r = −0.628) | (*P = <0.001, r = −0.843) |
| CaO$_2$ (mg·dL$^{-1}$) | 20.2 ± 1.1 | 20.1 ± 1.4 | 21.0 ± 2.8 |

Values are mean ± SD based on nine lowlanders and nine highlanders. BMI, body mass index; Hb, haemoglobin; Hct, haematocrit; cP, centipoise; PvO$_2$/PvCO$_2$, venous partial pressure of oxygen/carbon dioxide; HCO$_3^-$, bicarbonate; SpO$_2$, systemic arterial oxyhemoglobin saturation; CaO$_2$, arterial oxygen content. Comparisons performed using a combination of parametric tests (paired samples *t*-tests for lowlander comparison and independent samples *t*-tests for highlander comparisons). Non-parametric equivalents are also provided in brackets (Wilcoxon signed rank and Mann-Whitney *U* tests). *different (either within and/or between groups) compared to Lowlanders at Lifelong Sea-Level (lifelong-SL); †different (between groups) compared to Lowlanders at Chronic High-Altitude (chronic-HA). d, Cohen's (parametric) effect size; r, Wilcoxon (non-parametric) effect size.

centrifugation at 600 *g* (4°C) for 10 min. Serum, plasma and red blood cell samples were decanted into cryogenic vials (Nalgene Labware, Thermo Fisher Scientific, Waltham, MA, USA) and immediately flash-frozen in liquid nitrogen prior to transport to the UK (on dry ice). Samples were left to defrost at 37°C in the dark for 3 min before batch analysis.

**Haematology.** Venous blood was also directly assayed for haematocrit (Hct), haemoglobin (Hb), partial pressure of oxygen/carbon dioxide ($P_{vO_2}/P_{vCO_2}$), bicarbonate ($HCO_3^-$), pH and glucose using a commercially available blood gas analyser (ABL-90, Radiometer, Copenhagen, Denmark). Whole blood viscosity was measured in duplicate at a shear rate of 225 s$^{-1}$ at 37°C using a cone and

plate viscometer (DV2T Viscometer, Brookfield Amtek, Middleborough, MA, USA) (Tremblay et al., 2019).

**Free radicals.** The ascorbate free radical ($A^{\bullet-}$) was employed as a direct measure of systemic free radical formation with an increase in $A^{\bullet-}$ representing an increase in global, systemic flux of free radicals (Bailey et al., 2018). Exactly 1 mL of plasma was injected into a high-sensitivity multiple-bore sample cell (AquaX, Bruker Daltonics Inc., Billerica, MA, USA) housed within a $TM_{110}$ cavity of an electron paramagnetic resonance (EPR) spectrometer operating at X-band (9.87 GHz). Samples were recorded by cumulative signal averaging of 10 scans using the following instrument parameters: resolution, 1024 points; microwave power, 20 mW; modulation amplitude, 0.65 G; receiver gain, $2 \times 10^5$; time constant, 40.96 ms; sweep rate, 0.14 $G \cdot s^{-1}$; sweep width, 6 G; centre field, 3486 G. All spectra were filtered identically (moving average, 15 conversion points) using WINEPR software (version 2.11, Bruker, Karlsruhe, Germany) and the double integral of each doublet quantified using Origin 8 software (OriginLab Corp., Northampton, MA, USA). The intra- and inter-assay coefficients of variation (CVs) were both <5%.

**Nitric oxide metabolites.** Ozone-based chemiluminescence (Sievers NOA 280i, Analytix Ltd, Durham, UK) was employed to detect NO liberated from plasma and red blood cell (RBC) samples via chemical reagent cleavage (Bailey et al., 2017). This facilitated detection of plasma nitrite ($NO_2^-$) and *S*-nitrosothiols (RSNO) and RBC $NO_2^-$, *S*-nitrosohaemoglobin (SNO-Hb) and iron nitrosylhaemoglobin (HbNO). Total plasma NO was calculated as the sum of plasma $NO_2^-$ + RSNO and total RBC NO as RBC $NO_2^-$ + SNO-Hb + HbNO. Signal output (mV) was plotted against time (s) using Origin 8 software and smoothed using a 150-point averaging algorithm. The Peak Analysis package was used to calculate the area under the curve ($mV \cdot s^{-1}$) and subsequently converted to a concentration, using standard curves of known concentrations of sodium nitrite. The intra- and inter-assay CVs for all metabolites were <10%. All chemicals were of the highest available purity from Sigma–Aldrich (Gillingham, UK).

**NVU proteins.** A panel of serum biomarkers indirectly reflecting the structural integrity of the constituent components of the NVU were quantified using ultrasensitive analytical platforms as recently described (Bailey et al., 2022). S100B, a calcium-binding protein expressed and released predominantly by astrocytes and Schwann cells found at the perivascular brain space (Michetti et al., 2019), and glial fibrillary acidic protein (GFAP), an intermediate filament protein expressed

predominantly by astrocytes (Bignami et al., 1972), were employed as biomarkers of BBB permeability and glio-vascular damage. Neuron specific enolase (NSE), an intracytoplasmic glycolytic enzyme derived from neuronal cytoplasm and neuroendocrine cells (Pahlman et al., 1984), NF-L, a component of the axonal cytoskeleton expressed primarily in large-calibre myelinated subcortical axons (Friede & Samorajski, 1970), and ubiquitin carboxy-terminal hydrolase-L1 (UCH-L1), a neuron-specific cytoplasmic enzyme concentrated in dendrites (Thompson et al., 1983), were taken to reflect neuronal-axonal damage. Total (T)-tau, a microtubule-associated protein expressed predominantly in short cortical unmyelinated axons (Trojanowski et al., 1989), was employed as a surrogate for axonal degeneration. The underlying source(s) including extracranial contributions, biochemistry, detection and clinical interpretation of these biomarkers have recently been reviewed (Janigro et al., 2020). Automated high-sensitivity clinical grade ELISA (Liaison, DiaSorin, Saluggia, Italy) was used to measure S100B and NSE. The Neurology 4-Plex assay kit (Quanterix Corp., Lexington, MA, USA) (Rissin et al., 2010) was employed to measure GFAP, NF-L, UCH-L1 and T-tau proteins on a single molecule array (Simoa) HD-1 Analyzer (Quanterix Corp). Based on singulation of enzyme labelled immune-complex on paramagnetic beads, this digital ELISA is considered ~1200-fold more sensitive than conventional immunoassays (Wilson et al., 2016). All samples were analysed following (4-fold) dilution with the diluent provided in the kit (phosphate buffer with bovine serum and heterophilic blocker solution) to minimize matrix effects. The intra- and inter-assay CVs for all metabolites were <5%.

### Cardiopulmonary function

All cardiorespiratory variables were sampled continuously at 1000 Hz using an analogue-to-digital converter (Powerlab, 16/30; ADInstruments, Colorado Springs, CO, USA) and data were interfaced with LabChart (version 7.1), and analysed offline. Cardiopulmonary variables included mean arterial pressure (MAP), systolic blood pressure (SBP), diastolic blood pressure (DBP), heart rate (HR), stroke volume (SV), cardiac output ($\dot{Q}$), peripheral oxygen saturation ($S_{pO_2}$), minute ventilation ($\dot{V}_E$), and partial pressure of end-tidal oxygen ($P_{ETO_2}$) and carbon dioxide ($P_{ETCO_2}$). Beat-by-beat blood pressure was assessed using non-invasive finger photoplethysmography (Finometer PRO, Finapres Medical Systems, Amsterdam, Netherlands) and was calibrated each time prior to experimentation using the return-to-flow function. The Finometer blood pressure waveform was used to calculate MAP after calibrating values to the average of

two automated brachial blood pressure measurements (Life Source, A&D Medical, model UA767FAM, Ontario, Canada), taken over a 5 min resting baseline period. A three-lead electrocardiogram ((ECG) ADInstruments BioAmp ML132)) was used to assess HR. Respiratory gases (i.e. $CO_2$ and $O_2$) were sampled at the mouth and continuously recorded using a gas analyser (model ML206, ADInstruments), which was calibrated daily to atmospheric pressure. $P_{ETCO_2}$ and $P_{ETO_2}$, were calculated using peak detection analysis. Respiratory flow was measured with a pneumotachometer (model HR 800 L, Hans Rudolph, Shawnee, KS, USA). SV was estimated from the arterial blood pressure waveform, via finger photoplethysmography, by using the Model Flow method (Jellema et al., 1999). $\dot{Q}$ was derived mathematically as: HR × SV. $S_{pO_2}$ was measured continuously using pulse oximetry (Nonin 9550 Onyx II, Nonin Medical, Inc., Plymouth, MI, USA). Hb and $S_{pO_2}$ were used to calculate arterial oxygen content ($C_{aO_2}$) as follows:

$$C_{aO_2} \, (mL \cdot dL^{-1}) = 1.39 \times [Hb] \, (g \cdot dL^{-1}) \times S_{pO_2} \, (\%) \, /100$$

### Cerebrovascular function

**CBF.** Following 30 min of supine rest, the proximal segment of the right middle cerebral artery (MCA) and left posterior cerebral artery (PCA) was insonated bi-laterally using two 2 MHz-pulsed transcranial Doppler (TCD) ultrasound probes (2 MHz, TCD, Spencer Technologies, Seattle, WA, USA). Following standard searching techniques (Willie et al., 2011), the Doppler probes were secured over the middle trans-temporal window using a custom fit headband device (Spencer Technologies) to measure cerebral blood velocity in both the MCA (MCAv) and PCA (PCAv) simultaneously. The same experienced researcher reliably performed TCD for all participants. Diameter and blood velocity of the right internal carotid artery (ICA) and left vertebral artery (VA) were measured using a 10 MHz multi-frequency linear array duplex ultrasound (Terason uSmart 3300, Teratech, Burlington, MA, USA). Specifically, B-mode imaging was used to measure arterial diameter, while simultaneous live pulse-wave mode was used to concurrently measure peak blood velocity. Diameter and velocity of the ICA were measured at least 1.5 cm distal to the common carotid bifurcation to eliminate recordings of turbulent and retrograde flow and non-uniform shear. VA blood velocity and diameter were measured between C4 and C5 or C5 and C6. Care was taken to ensure that the insonation angle (60°) was unchanged throughout each test. All recordings were made in accordance with published technical recommendations (Thomas et al., 2015). Recordings were screen-captured and stored as video files for offline analysis. This analysis involved concurrent determination of arterial diameter and peak blood velocity at 30 Hz, using customized edge detection and wall tracking software (BloodFlow Analysis, version 5.1) designed to mitigate observer bias (Woodman et al., 2001). Volumetric blood flow ($\dot{Q}$) was calculated as:

$$ICA_{\dot{Q}} \text{ or } VA_{\dot{Q}} (mL \cdot min^{-1}) = \left( \frac{1}{2} \text{Peak Envelope Velocity} \right)$$
$$\times \left( \pi \left( \frac{1}{2} \text{ Diameter} \right)^2 \right) \times 60$$

Notwithstanding potential limitations associated with contralateral asymmetry (Mysior & Stefanczyk, 2007), global CBF (gCBF) was calculated as:

$$gCBF \, (mL \cdot min^{-1}) = \left( ICA_{\dot{Q}} + VA_{\dot{Q}} \right) \times 2$$

Cerebrovascular conductance index (CVCi) was calculated by dividing CBV or CBF by MAP and the cerebrovascular resistance index (CVRi) derived as MAP divided by CBV or CBF. MCA and PCA pulsatility indices (PIs) were calculated as the difference between respective systolic and diastolic CBVs, divided by mean CBVs (Gosling & King, 1974).

**Dynamic cerebral autoregulatory capacity.** Following 30 min of supine rest, a 5-min segment of MAP, MCAv and PCAv was obtained for spectral analysis of spontaneous oscillations to assess regional dynamic cerebral auto-regulatory capacity (dCA) via transfer function analysis (TFA) (Zhang et al., 1998). Beat-to-beat MAP, MCAv and PCAv signals were calculated across each cardiac cycle, linearly interpolated and resampled at 4 Hz in accordance with formal recommendations of the Cerebral Autoregulation Research Network (Claassen et al., 2016). Spontaneous MAP and CBV power spectrum density and the mean value of TFA coherence, gain and phase of the spontaneous oscillations were band averaged across the very low frequency (VLF; 0.02–0.07 Hz, 50–14.3 s cycles) and low frequency (LF; 0.07–0.2 Hz, 14.3–5 s cycles) ranges where cerebral autoregulation is considered to be most operant (Zhang et al., 1998). To ensure that robust phase and gain estimates were entered for subsequent analysis, we averaged only those gain and phase (positive to eliminate wrap-around) values where the corresponding coherence was ≥0.5. Accordingly, an increase in gain and reduction in phase were taken to reflect impaired dCA indicative of a more pressure-passive relationship between MAP and MCAv/PCAv (Bailey et al., 2019).

**Neurovascular coupling.** The NVC assessment measured the activation of the visual cortex to allow for selective changes in PCAv. TCD ultrasound was used to measure PCAv during repeated dark (eyes closed) and light (eyes open + visual stimulation) activation. Following the

5 min resting baseline, five repeated cycles of alternating 30 s of eyes-closed, and then 30 s eyes-open whilst looking at a red-dot in the centre of a flashing screen (alternative checkboard squares) were completed (Hoiland et al., 2020). Throughout the assessment, that the participant's eyes were closed and open during each 30 s period was confirmed by a researcher. If an artefact occurred within a cycle, another cycle was added and used to replace the aberrant cycle. A period of 5 min, before the assessment of NVC, was recorded for resting baseline values, and a 25 s average during the eyes-closed cycle was used to calculate the change from baseline during NVC. Absolute and peak changes during the activation stage (eyes-open + visual stimulation) of NVC were calculated. All raw signals were extracted for analysis using custom software developed in MATLAB (Phillips et al., 2016).

**Cerebrovascular reactivity to carbon dioxide.** A custom-designed dynamic end-tidal forcing (DEF) system (Tymko et al., 2015) was used to control $P_{ETCO_2}$, independent of alterations in pulmonary ventilation, for the assessment of $CVR_{CO_2}$. By utilizing predictive feedforward algorithms, feedback information and error reduction algorithms, $P_{ETCO_2}$ was controlled by adjusting the necessary fraction of inspired carbon dioxide ($F_{ICO_2}$) on a breath-by-breath basis. Participants were paced at 20 breaths $min^{-1}$ through verbal coaching and visual feedback while end-tidal values were maintained (i.e. did not change) for 2 min, to improve the ability of the end-tidal forcing systems to abruptly manipulate $P_{ETCO_2}$ in a controlled fashion. $P_{ETCO_2}$ was shortly ($\sim$5 s) increased to +9 mmHg above baseline for 3 min. $CVR_{CO_2}$ was calculated as the percentage increase in CBV ($MCA_V$/$PCA_V$) and CBF ($ICA_{\dot{Q}}$/$VA_{\dot{Q}}$) from baseline per 1 mmHg increase in $P_{ETCO_2}$ (Hoiland et al., 2022).

### Cognitive function

Cognitive function was examined using a computerized identification test (IDT) and two-back version of the *N*-back task (TBT) (Cogstate, New Haven, CT, USA). The identification test was used to assess attention and required participants to decide whether a playing card that is turned over was red by indicating 'yes'. The two-back test was used to assess working memory and required participants to identify whether the card they are being shown was the same as the card shown before the previous card by indicating 'yes'. Throughout both tests, participants were encouraged to work as quickly and accurately as possible. Participants completed both tasks in the same order whilst supine, using a standard mouse and a computer screen that was placed 60 cm above the participant's head. A right click of the mouse indicated

'yes'. Additionally, the Montreal Cognitive Assessment (MoCA) was employed to examine visuospatial/executive, naming, memory, attention, language, abstraction, delayed recall and orientation with each section having a designated point value with a maximum score of 30 points (Nasreddine et al., 2005) and the test scored by the number of correct answers given. Education status was recorded and those participants with 12 years or less of education had a point added to their total MoCA score (Bailey et al., 2019).

### Statistical analysis

Data were analysed using the Statistics Package for Social Scientists (IBM SPSS Statistics Version 29.0, IMB Corp., Armonk, NY, USA). (version 29.0; IBM Corp., Armonk, NY, USA). While repeated Shapiro–Wilk W tests confirmed that all datasets were normally distributed (all $P > 0.050$), we complimented our parametric analyses with non-parametric equivalents. Lowlander datasets were analysed using paired samples Student's *t*-tests and Wilcoxon signed rank tests. Highlander data were compared to lowlanders at SL (lifelong SL) and HA (chronic HA) using an independent-samples *t*-tests and the Mann–Whitney U tests. Comparisons between highlanders (lifelong HA) and lowlanders at SL (lifelong SL) provides insight into both populations in their natural residence. It was important to also compare acclimatized lowlanders (chronic HA) with highlanders to determine the difference between biological and evolutionary acclimatization to HA. Effect sizes were calculated using Cohen's *d* and Wilcoxon *r* (non-parametric equivalent) tests. Significance was established at $P < 0.05$ and data expressed as means $\pm$ standard deviation (SD).

## Results

### Parametric *vs.* non-parametric outcomes

With the exception of $A^{\bullet -}$ and S100B, all qualitative outcomes persisted following complimentary non-parametric analyses (see below).

### Demographics

By design, both groups were prospectively matched for sex (all male), age ($P = 0.827$, $d = 0.105$) and select anthropometric variables (Mass: $P = 0.470$, $d = 0.349$; BMI: $P = 0.412$, $d = 0.404$) with the exception of shorter stature typical of highlanders (Table 1). Highlanders were also characterized by a lower educational status compared to lowlanders (13 $\pm$ 3 *vs.* 20 $\pm$ 2 years, $P < 0.001$, $d = 3.119$).

## Molecular function

**OXNOS.** In lowlanders, chronic HA decreased $A^{\bullet-}$ ($P = 0.034$, $d = -0.852$, borderline significant following non-parametric analysis: $P = 0.051$, $r = -0.461$). This was associated with a reciprocal elevation in total plasma ($NO_2^- + RSNO$) and RBC NO ($NO_2^- + SNO\text{-}Hb + HbNO$) bioavailability ($P = 0.003$, $d = 1.385$ and $P = 0.026$, $d = 0.905$, respectively), primarily due to an elevation in plasma $NO_2^-$ ($P = 0.003$, $d = 1.385$), plasma RSNO ($P = 0.006$, $d = 1.237$), RBC $NO_2^-$ ($P = 0.0495$, $d = 0.771$) and HbNO ($P = 0.010$, $d = 1.108$, Fig. 1). Total plasma and total RBC NO bioavailability were consistently elevated in highlanders when compared to lowlanders at lifelong SL ($P = 0.005$, $d = 1.549$ and $P = 0.003$, $d = 1.681$, respectively). No differences were observed for $A^{\bullet-}$ when comparing highlanders to lowlanders at lifelong SL ($P = 0.672$, $d = -0.203$) or following chronic HA ($P = 0.157$, $d = 0.701$, Fig. 1).

**NVU integrity.** Chronic HA was associated with an elevation in NF-L ($P = 0.030$, $d = 0.877$) and reduction in T-tau ($P = 0.006$, $d = -1.230$) compared to lowlanders at SL, whereas no differences were observed for S100B, NSE, GFAP or UCH-L1. Highlanders exhibited consistently lower S100B ($P = 0.046$, $d = -1.019$, albeit significance was lost for non-parametric analyses: $P = 0.094$, $r = -0.414$), UCH-L1 ($P = 0.021$, $d = -1.203$) and T-tau ($P < 0.001$, $d = -2.676$) compared to lowlanders at SL. Furthermore, S100B ($P = 0.018$, $d = -1.244$), NF-L ($P = 0.037$, $d = -1.075$) and T-tau ($P < 0.001$, $d = -3.894$) were also lower in highlanders compared to lowlanders at chronic HA. No between-group differences were observed in NSE or GFAP (all $P > 0.050$) (Fig. 2).

## Cardiopulmonary function

Despite the reduction in $S_{pO_2}$ in lowlanders ($P < 0.001$, $d = -4.177$), chronic HA increased Hct ($P < 0.001$, $d = 3.075$), Hb ($P < 0.001$, $d = 3.072$) and blood viscosity ($P < 0.001$, $d = 1.975$), collectively restoring $C_{aO_2}$ to baseline SL (normoxic) values ($P = 0.697$, $d = 0.134$) (Table 1). Compared to SL, chronic HA also decreased $HCO_3^-$ ($P = 0.001$, $d = -1.802$) and increased blood pH ($P < 0.001$, $d = 2.021$) in lowlanders. While highlanders were more polycythaemic ($P = 0.016$, $d = 1.267$) and acidic ($P = 0.001$, $d = 2.21$), they were equally hypoxaemic ($P = 0.205$, $d = 0.623$) albeit slightly less hypocapnic ($P < 0.001$, $d = -2.476$) and hypertensive ($P = 0.032$, $d = -1.106$) compared to lowlanders following chronic HA (Table 1 and 2).

## Cerebrovascular function

**CBF and substrate delivery.** Following chronic HA, lowlanders presented with reduced $PCA_V$ ($P = 0.004$, $d = -1.327$), whereas $MCA_V$, $ICA_{\dot{Q}}$, $VA_{\dot{Q}}$, global cerebral blood flow (gCBF), global cerebral delivery of glucose ($gCD_{Glu}$) and global cerebral delivery of oxygen ($gCD_{O_2}$) remained unchanged (all $P > 0.050$). No between-group differences were observed for $MCA_V$, $PCA_V$, $ICA_{\dot{Q}}$, $VA_{\dot{Q}}$, gCBF, $gCD_{O_2}$, $gCD_{Glu}$ (all $P > 0.050$, Fig. 3). Chronic HA in lowlanders also decreased MCA CVCi ($P = 0.011$, $d = -1.099$) and PCA CVCi ($P = 0.001$, $d = -1.819$) and increased MCA CVRi ($P = 0.010$, $d = 1.127$) and PCA CVRi ($P < 0.001$, $d = 2.061$). Highlanders exhibited lower MCA PI and PCA PI compared to lowlanders at SL ($P < 0.001$, $d = -3.428$ and $P < 0.001$, $d = -3.481$, respectively) and lowlanders at chronic HA ($P < 0.001$, $d = -2.037$ and $P < 0.001$, $d = -2.226$, respectively, Table 2).

**dCA.** Compared to SL, chronic HA increased MAP VLF power spectral density (PSD; $P = 0.031$, $d = 0.868$), MCA VLF coherence ($P = 0.043$, $d = 0.803$), MCA LF coherence ($P = 0.044$, $d = 0.795$) and PCA LF coherence ($P < 0.001$, $d = 1.940$) and decreased PCA VLF gain ($P = 0.039$, $d = -0.822$) and PCA LF gain ($P = 0.022$, $d = -0.945$). Highlanders exhibited lower MCA LF PSD ($P = 0.049$, $d = -1.087$), PCA LF PSD ($P = 0.023$, $d = -1.227$), MCA LF phase ($P = 0.020$, $d = -1.391$), MCA LF gain ($P = 0.029$, $d = -1.291$) and PCA LF gain ($P = 0.017$, $d = -1.255$) compared to lowlanders at SL and lower MCA LF PSD ($P = 0.049$, $d = -1.087$), PCA VLF PSD ($P = 0.050$, $d = -1.038$), PCA LF PSD ($P = 0.007$, $d = -1.512$), MCA VLF coherence ($P = 0.010$, $d = -1.498$), MCA LF coherence ($P = 0.001$, $d = -2.192$) and PCA VLF coherence ($P = 0.037$, $d = -1.071$) compared to lowlanders following chronic HA (Fig. 4).

**$CVR_{CO2}$.** As anticipated, MAP increased during the $CVR_{CO_2}$ challenge, but this was comparable across conditions and groups (all $P > 0.050$). In lowlanders, MCA $CVR_{CO_2}$ and ICA $CVR_{CO_2}$ were higher following chronic HA ($P = 0.036$, $d = 0.836$ and $P = 0.042$, $d = 0.807$, respectively). Highlanders presented with elevated ICA $CVR_{CO_2}$, compared to lowlanders at SL. No differences were observed for PCA $CVR_{CO_2}$ or VA $CVR_{CO_2}$ between conditions or groups (Fig. 5). In lowlanders, chronic HA failed to alter regional CVCi during the $CVR_{CO_2}$ challenge (CBF response thus 'normalized' to both (hypercapnia-induced elevations in $P_{ETCO_2}$ and MAP; all comparisons $P > 0.05$). In contrast, ICA CVCi was selectively elevated in the highlanders compared to lowlanders at SL ($P = 0.046$, $d = 1.087$).

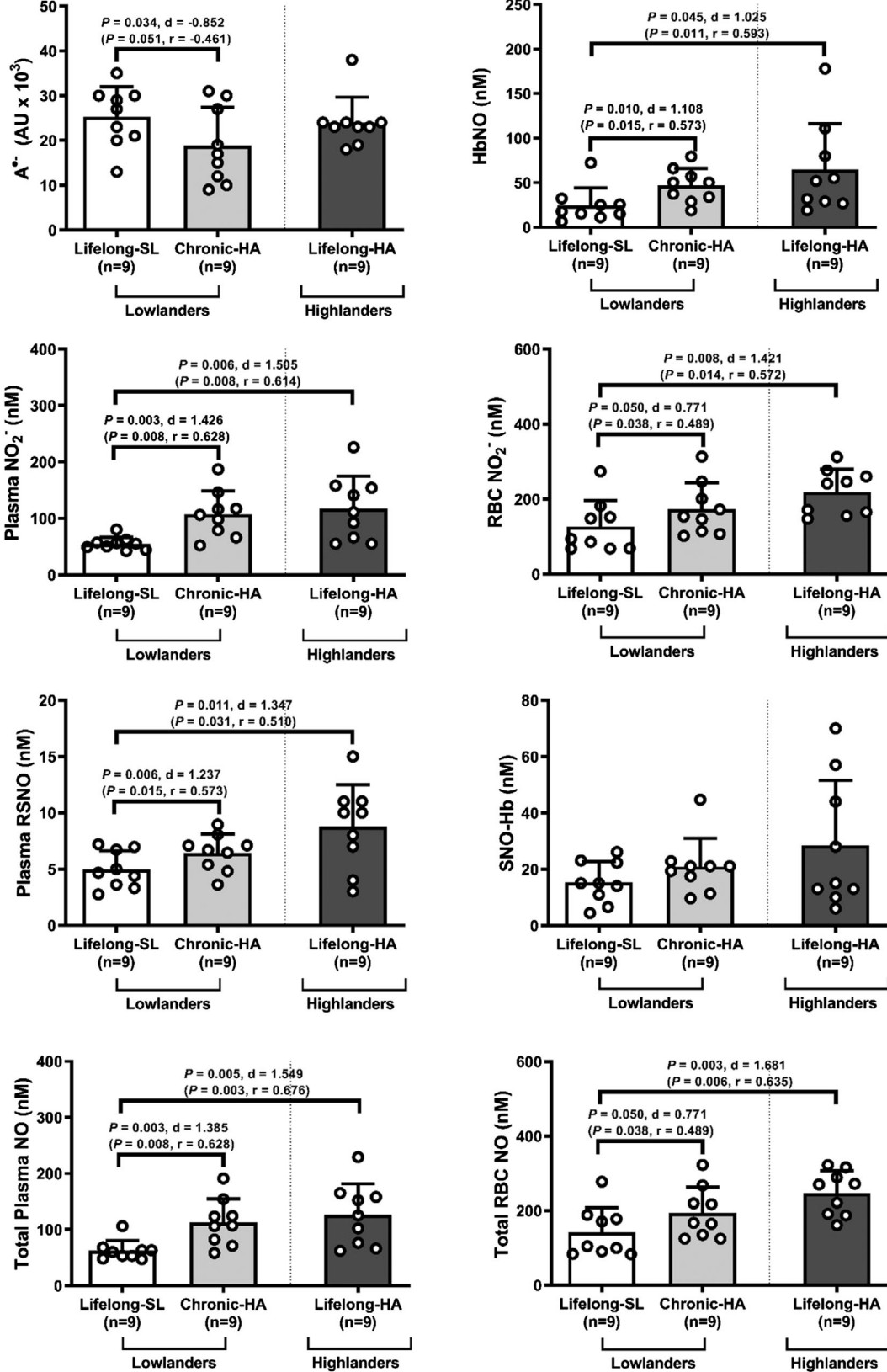

**Figure 1. Systemic concentrations of free radicals and nitric oxide metabolites**

**NVC.** Compared to SL, lowlanders following chronic HA presented with a greater increase in percentage peak and average $PCA_V$ in response to visual stimulation ($P = 0.005$, $d = 1.616$ and $P = 0.017$, $d = 1.616$, respectively). Highlanders demonstrated intact and comparable NVC responses to lowlanders following chronic HA (all $P > 0.050$, Fig. 6).

## Cognitive function

Cognitive function is represented by the accuracy of the identification test (IDT) and two-back test (TBT), and score from the Montreal Cognitive Assessment (MoCA). In lowlanders, chronic HA decreased IDT accuracy ($P = 0.023$, $d = -0.932$) whereas no changes were observed in either TBT accuracy or MoCA score. MoCA scores were comparable between highlanders and lowlanders (both lifelong SL and chronic HA), although the latter was borderline significant and retrospective analyses revealed inadequate statistical power (parametric: $P = 0.056$, $d = 0.973$, $1 - \beta = 0.494$ and non-parametric: $P = 0.113$, $r = 0.391$ *vs.* lifelong HA). Despite rigorous medical screening, one lowlander (both at lifelong SL and chronic HA) and one highlander presented with a MoCA score of 24–25 points (Fig. 7), indicative of mild cognitive decline according to established guidelines (Nasreddine et al., 2005).

## Discussion

The Altiplano has been occupied since close to the terminal Pleistocene ~12,000 years ago, placing its inhabitants as one of the most ancient populations permanently living at HA (Aldenderfer, 2003; Beall, 2006; Haas et al., 2017). Having evolved under the selective pressures imposed by severe lifelong hypoxia, Andean highlanders exhibit distinct cardiopulmonary adaptations that collectively improve systemic $O_2$ transport (Beall, 2007). Extending these observations to the cerebral circulation, the novel integrated translational approach employed by the present study is the first to provide evidence for NVU adaptation in highlanders, taking the form of improved molecular-haemodynamic biomarkers underpinning NVU structure and function. Compared to lowlanders at SL, highlanders were characterized by elevated basal NO bioavailability, improved dCA and cerebrovascular reactivity and lower systemic concentrations of S100B, UCHL-1 and T-tau in the face of preserved cerebral bioenergetics, NVC and cognition. Collectively, these novel findings tentatively suggest that these integrated adaptations in highlanders may reflect a neuroprotective phenotype to cope with the lifelong stress of HA hypoxia.

## Molecular function

The present study utilized the EPR spectroscopic detection of $A^{\bullet-}$ as a direct measure of global free radical formation (Bailey et al., 2018). Given that the concentration of ascorbate in the human systemic circulation is greater than that of any other oxidizing free radical, combined with the low one-electron reduction potential for the $A^{\bullet-}$/ascorbate monanion ($AH^-$) couple ($E° = 282$ mV) (Williams & Yandell, 1982), any oxidizing species ($R^\bullet$) generated locally within the systemic circulation will result in the one-electron oxidation of ascorbate to form the distinctive EPR-detectable $A^{\bullet-}$ 'doublet' ($R^\bullet + AH^- \rightarrow A^{\bullet-} + R–H$) (Buettner & Jurkiewicz, 1993).

Chronic HA decreased $A^{\bullet-}$ in lowlanders, coinciding with an elevation in plasma and RBC NO bioavailability, which was comparable to that of highlanders. Although in contrast to prior findings albeit in Bolivian highlanders (Bailey et al., 2013; Bailey et al., 2019), these findings are supported by earlier observations of elevated basal plasma $NO_2^-$ and RSNO concentrations in lowlanders following chronic HA (Levett et al., 2011) and in native Tibetan highlanders (Erzurum et al., 2007).

The elevation in vascular NO bioavailability observed following chronic HA in lowlanders and lifelong HA in highlanders is likely mediated by an increase in the deoxyHb-mediated reduction of $NO_2^-$ to NO and elevated *de novo* synthesis by NO synthase isoforms, converting L-arginine to NO and L-citrulline (Beall et al., 2012; Bigham et al., 2010). While thermodynamically plausible (Bailey, Taudorf et al., 2009), the possibility that the lower $A^{\bullet-}$ observed following chronic HA in lowlanders was related to enhanced NO scavenging (Singh et al., 2012) is unlikely given that the elevated basal $A^{\bullet-}$ in highlanders persisted in the face of an equivalent elevation in vascular NO bioavailability. These findings further emphasize elevated vascular NO bioavailability to be an adaptive and integral response to sustained

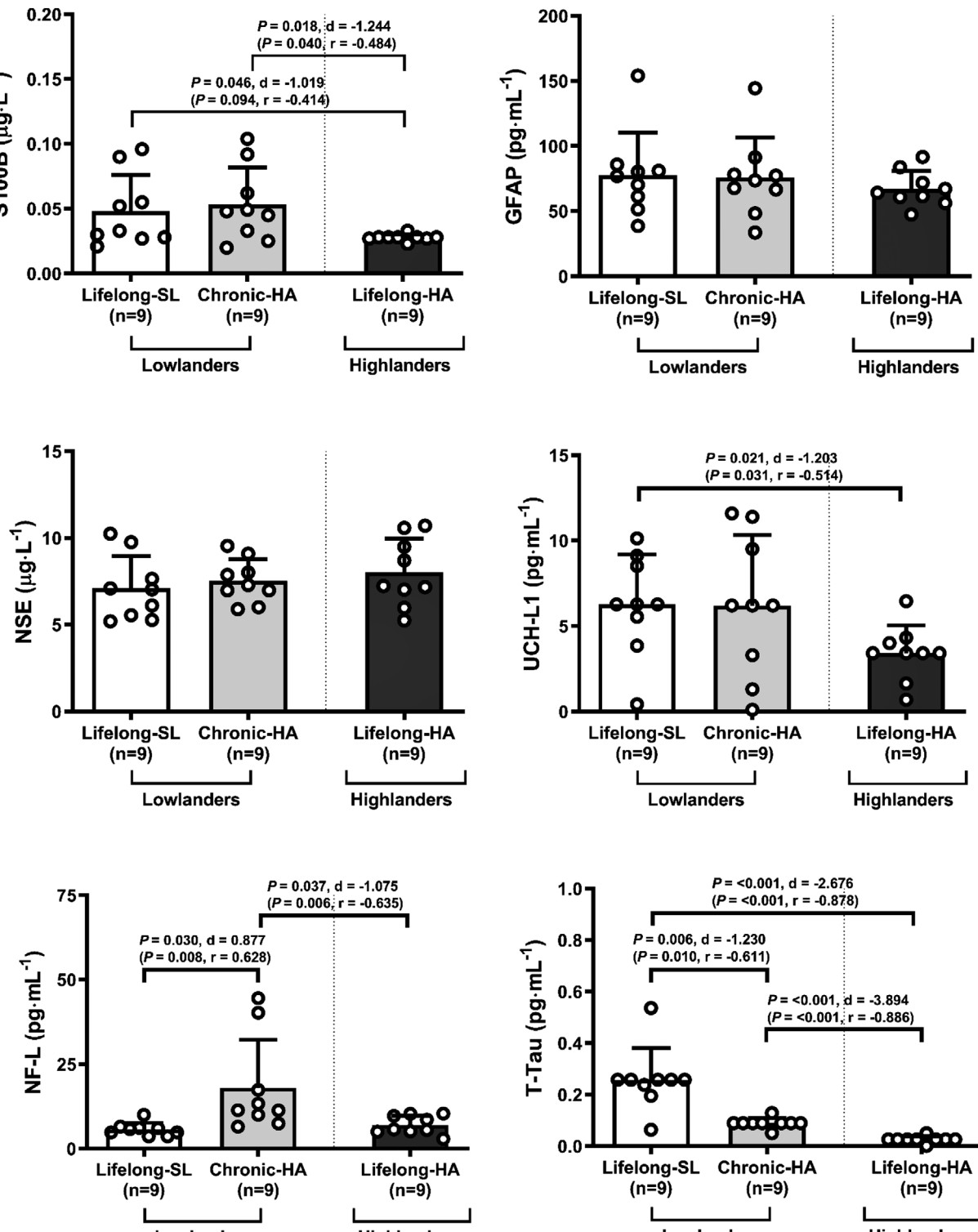

**Figure 2. Systemic concentrations of brain-specific proteins reflecting structural integrity injury of the neurovascular unit**
Values are means ± SD. GFAP, glial fibrillary acidic protein; HA, high altitude; NF-L, neurofilament light chain; NSE, neuron-specific enolase; SL, sea level; UCH-L1, ubiquitin carboxy-terminal hydrolase-L1. Comparisons performed using a combination of parametric tests (paired samples *t*-tests for lowlander comparison and independent samples *t*-tests for highlander comparisons). Non-parametric equivalents also provided in brackets (Wilcoxon signed rank and Mann-Whitney *U* tests). *d*, Cohen's (parametric) and *r*, Wilcoxon's (non-parametric) effect sizes.

**Table 2. Cardiopulmonary and cerebrovascular data**

| Group: | Lowlanders (n = 9) | | Highlanders (n = 9) |
|---|---|---|---|
| Exposure-Location: | Lifelong-SL | Chronic-HA | Lifelong-HA |
| **Cardiopulmonary** | | | |
| MAP (mmHg) | 83 ± 9 | 94 ± 8 <br> *P = 0.001, d = 1.856 <br> (*P = 0.008, r = 0.628) | 85 ± 8 <br> †P = 0.032, d = −1.106 <br> (†P = 0.031, r = −0.510) |
| HR (b·min$^{-1}$) | 57 ± 10 | 72 ± 11 <br> *P = 0.009, d = 1.139 <br> (*P = 0.008, r = 0.628) | 66 ± 10 |
| SV (mL) | 111 ± 26 | 97 ± 27 | 86 ± 15 <br> *P = 0.025, d = −1.168 <br> (*P = 0.031, r = −0.510) |
| $\dot{Q}$ (L·min$^{-1}$) | 6.2 ± 1.5 | 6.9 ± 1.8 | 5.6 ± 1.0 |
| $\dot{V}_E$ (L·min$^{-1}$) | 22 ± 10 | 16 ± 9 | 24 ± 7 <br> †P = 0.044, d = 1.032 <br> (†P = 0.040, r = 0.489) |
| $P_{ET}O_2$ (mmHg) | 91 ± 4 | 50 ± 3 <br> *P = <0.001, d = −7.256 <br> (*P = 0.008, r = −0.628) | 47 ± 2 <br> *P = <0.001, d = −13.286 <br> (*P = <0.001, r = −0.843) <br> †P = 0.011, d = −1.362 <br> (†P = 0.011, r = −0.593) |
| $P_{ET}CO_2$ (mmHg) | 43 ± 3 | 29 ± 2 <br> *P = <0.001, d = −4.131 <br> (*P = 0.008, r = −0.628) | 33 ± 2 <br> *P = <0.001, d = 3.942 <br> (*P = <0.001, r = 0.843) <br> †P = <0.001, d = 2.476 <br> (†P = <0.001, r = 0.780) |
| **Cerebrovascular** | | | |
| ICA CVCi (mL·min$^{-1}$·mmHg$^{-1}$) | 3.25 ± 0.67 | 2.85 ± 0.47 | 2.86 ± 0.66 |
| ICA CVRi (mmHg·mL$^{-1}$·min$^{-1}$) | 0.32 ± 0.07 | 0.36 ± 0.06 | 0.37 ± 0.09 |
| VA CVCi (mL·min$^{-1}$·mmHg$^{-1}$) | 1.19 ± 0.42 | 0.94 ± 0.27 | 0.89 ± 0.63 |
| VA CVRi (mmHg·mL$^{-1}$·min$^{-1}$) | 0.95 ± 0.35 | 1.16 ± 0.41 | 1.51 ± 0.82 |
| MCA CVCi (cm·s$^{-1}$·mmHg$^{-1}$) | 0.78 ± 0.19 | 0.62 ± 0.12 <br> *P = 0.011, d = −1.099 <br> (*P = 0.015, r = −0.573) | 0.71 ± 0.09 <br> †P = 0.102, d = 0.818 <br> (†P = 0.031, r = 0.500) |
| MCA CVRi (mmHg·cm$^{-1}$·s$^{-1}$) | 1.34 ± 0.29 | 1.65 ± 0.29 <br> *P = 0.010, d = −1.127 <br> (*P = 0.015, r = −0.573) | 1.43 ± 0.18 <br> †P = 0.069, d = −0.918 <br> (†P = 0.031, r = −0.510) |
| MCA PI (AU) | 1.08 ± 0.12 | 0.97 ± 0.13 <br> *P = 0.062, d = −0.724 <br> (*P = 0.038, r = −0.489) | 0.76 ± 0.06 <br> *P = <0.001, d = −3.428 <br> (*P = <0.001, r = −0.843) <br> †P = <0.001, d = −2.037 <br> (†P = 0.002, r = −0.688) |
| PCA CVCi (cm·s$^{-1}$·mmHg$^{-1}$) | 0.54 ± 0.11 | 0.40 ± 0.05 <br> *P = 0.001, d = −1.819 <br> (*P = 0.011, r = −0.618) | 0.44 ± 0.15 |
| PCA CVRi (mmHg·cm$^{-1}$·s$^{-1}$) | 1.92 ± 0.44 | 2.54 ± 0.29 <br> *P = <0.001, d = 2.061 <br> (*P = 0.011, r = 0.618) | 2.54 ± 1.02 |
| PCA PI (AU) | 1.10 ± 0.13 | 1.00 ± 0.16 | 0.72 ± 0.08 <br> *P = <0.001, d = −3.481 <br> (*P = <0.001, r = −0.840) <br> †P = <0.001, d = 2.226 <br> (†P = <0.001, r = 0.793) |

Values are mean ± SD based on nine lowlanders and nine highlanders. MAP, mean arterial blood pressure; HR, heart rate; SV, stroke volume; $\dot{Q}$, cardiac output; $\dot{V}_E$, minute ventilation; $P_{ET}O_2$/$P_{ET}CO_2$, end-tidal partial pressure of $O_2$/$CO_2$; MCA/PCA, middle/posterior

**Table 2. (Continued)**

cerebral artery; ICA/VA internal cerebral/vertebral artery; CVCi, cerebrovascular conductance index; CVRi, cerebrovascular resistance index; PI, pulsatility index. Comparisons performed using a combination of parametric tests (paired samples *t*-tests for lowlander comparison and independent samples *t*-tests for highlander comparisons). Non-parametric equivalents are also provided in brackets (Wilcoxon signed rank and Mann-Whitney *U* tests). *different (either within and/or between groups) compared to Lowlanders at Life-long Sea-Level (lifelong-SL); †different (between groups) compared to Lowlanders at Chronic High-Altitude (chronic-HA). d, Cohen's (parametric) effect size; r, Wilcoxon (non-parametric) effect size.

hypoxia and perhaps not simply the result of evolutionary selection.

The inclusion of clinically relevant protein biomarkers specific to the NVU provides unique insight into the structural integrity of the NVU, with a specific focus on the BBB. The appearance of the astrocytic protein S100B, a calcium-binding protein with a molecular mass of 20 kDa in the systemic circulation, is an established surrogate for increased BBB permeability (Janigro et al., 2020) and correlates directly with the extent

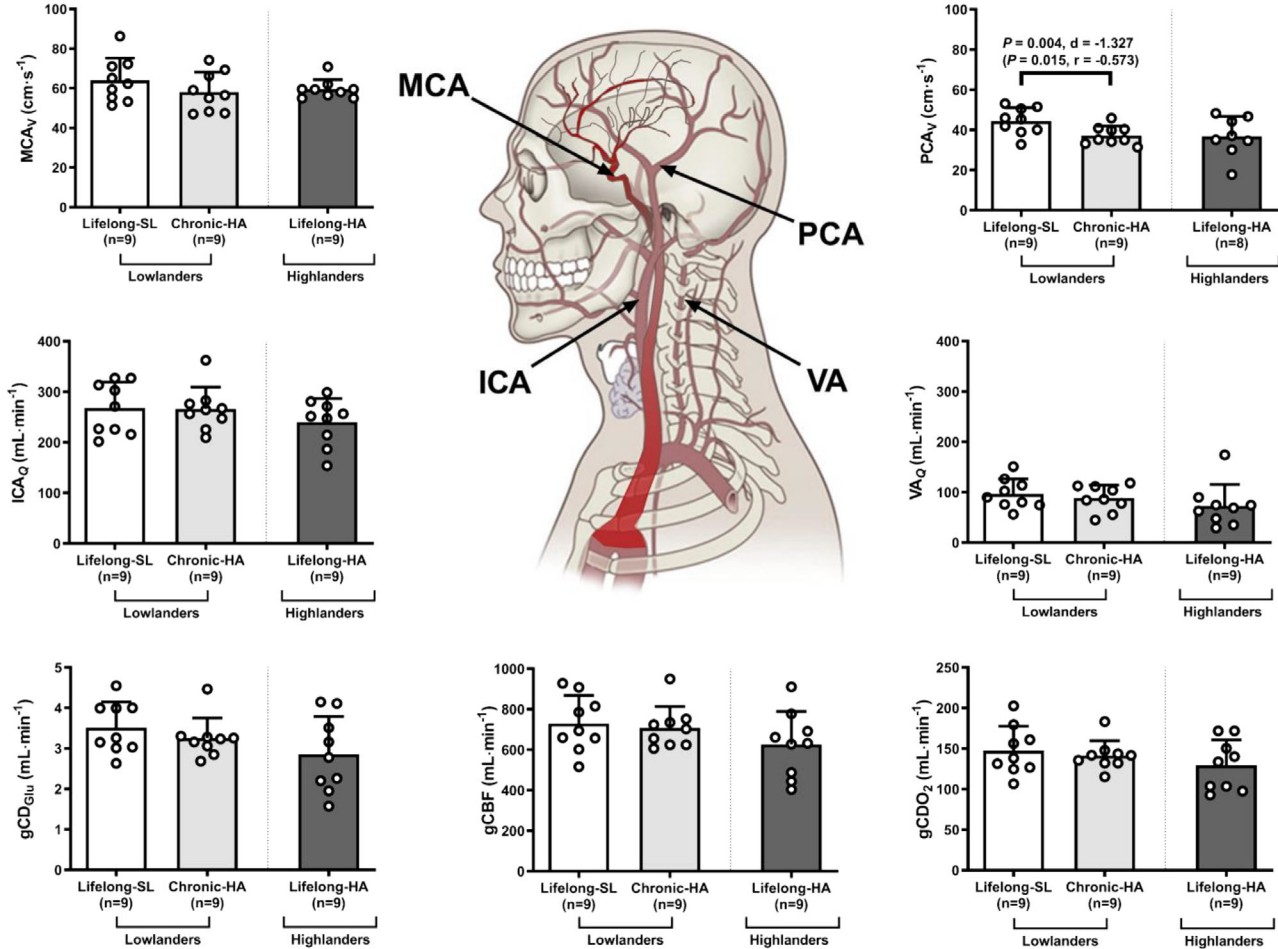

**Figure 3. Cerebral blood flow and substrate delivery**
Values are means ± SD. gCBF, global cerebral blood flow; gCD_Glu, global cerebral delivery of glucose; gCD_O2, global cerebral delivery of oxygen; HA, high altitude; ICA, internal carotid artery; $ICA_{\dot{Q}}$, internal carotid artery blood flow; MCA, middle cerebral artery; MCA_V, middle cerebral artery velocity; PCA, posterior cerebral artery; PCA_V, posterior cerebral artery velocity; SL, sea level; VA, vertebral artery; $VA_{\dot{Q}}$, vertebral artery blood flow. Comparisons performed using a combination of parametric tests (paired samples *t*-tests for lowlander comparison and independent samples *t*-tests for highlander comparisons). Non-parametric equivalents are also provided in brackets (Wilcoxon signed rank and Mann-Whitney *U* tests). d, Cohen's (parametric) and r, Wilcoxon's (non-parametric) effect sizes. [Colour figure can be viewed at wileyonlinelibrary.com]

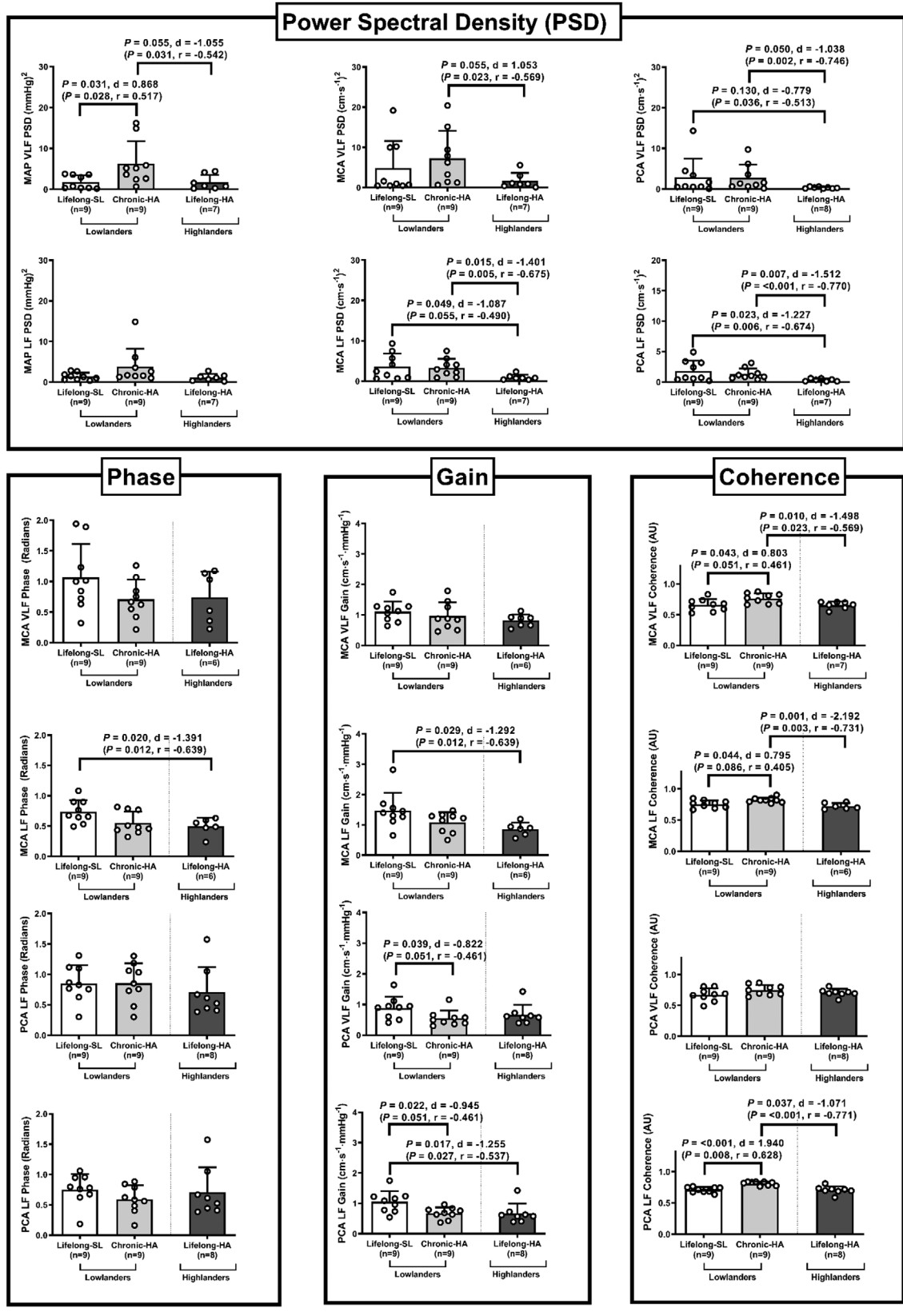

**Figure 4. Dynamic cerebral autoregulation**

Values are means ± SD based on 9 lowlanders and 9 highlanders. MCA, middle cerebral artery; PCA, posterior cerebral artery; PSD, power spectral density; SL, sea level; VLF, very low frequency. Comparisons performed using a combination of parametric tests (paired samples *t*-tests for lowlander comparison and independent samples *t*-tests for highlander comparisons). Non-parametric equivalents are also provided in brackets (Wilcoxon signed rank and Mann-Whitney *U* tests). *d*, Cohen's (parametric) and *r*, Wilcoxon's (non-parametric) effect sizes.

and temporal sequence of BBB opening confirmed by contrast-enhanced MRI (Kanner et al., 2003). Furthermore, elevated concentrations of NF-L, T-tau, NSE, UCHL-1 and GFAP in the systemic circulation is taken to reflect structural damage to neuronal and axonal cells (Cheng et al., 2014; Mattsson et al., 2017; Takala et al., 2016).

In the present study, highlanders exhibited lower basal concentrations of S100B, NF-L and T-tau compared to acclimatized lowlanders who presented with molecular evidence of axonal injury and who are typically more prone to hypoxia-induced BBB disruption, neuronal/gliovascular damage and corresponding neurological sequelae associated with HA illness (Bailey, Bartsch et al., 2009). The further suppression in T-tau in the highlanders is intriguing and may reflect lower neuronal secretion and/or selective uptake to optimize microtubule integrity. These findings are clinically relevant given the roles of NF-L and T-tau as prognostic biomarkers of global and domain-specific cognitive decline and neuroimaging markers of cortical thickness, hippocampal volume, white matter integrity

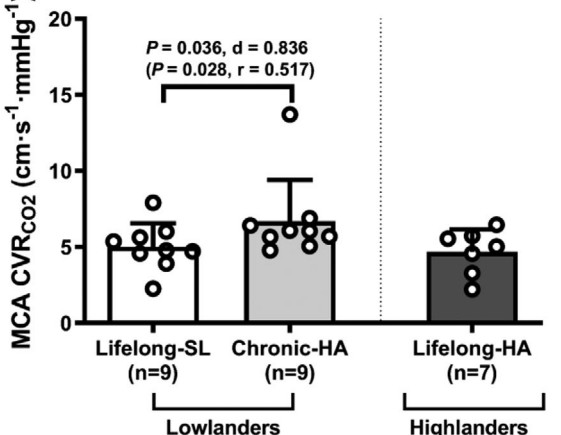

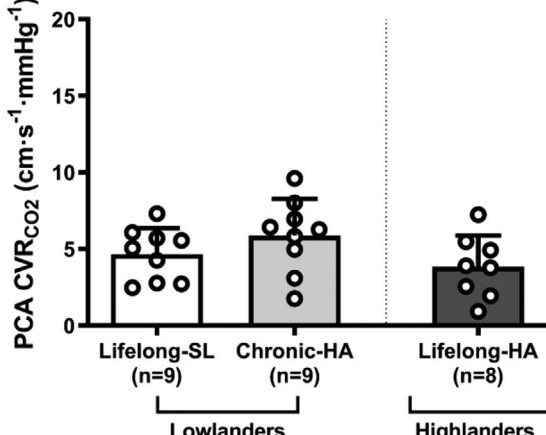

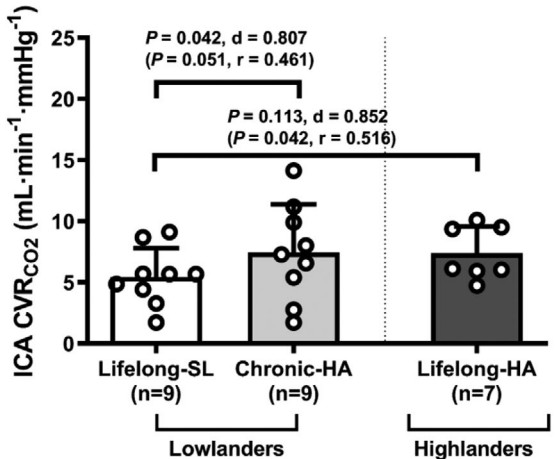

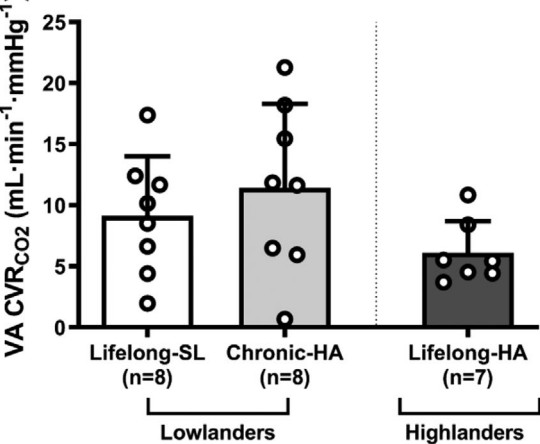

**Figure 5. Cerebrovascular reactivity to carbon dioxide**
Values are means ± SD based on 9 lowlanders and 9 highlanders. MCA, middle cerebral artery; PCA, posterior cerebral artery; SL, sea level; VA, vertebral artery. Comparisons performed using a combination of parametric tests (paired samples *t*-tests for lowlander comparison and independent samples *t*-tests for highlander comparisons). Non-parametric equivalents are also provided in brackets (Wilcoxon signed rank and Mann-Whitney *U* tests). *d*, Cohen's (parametric) and *r*, Wilcoxon's (non-parametric) effect sizes.

and white matter hyperintensity volume in patients without dementia (Marks et al., 2021).

However, it is important to acknowledge potential (unlikely, albeit unquantified) contributions from extracranial sources, including adipose tissue, skeletal tissue and cardiac glial cell release for S100B, neuromuscular junction for UCH-L1 and erythrocytes for NSE (Janigro et al., 2020). Caveats acknowledged, these findings provide preliminary evidence of evolutionary NVU adaptation to HA indicating that the NVU is structurally tighter or more intact, of which biological adaptation during acclimatization to chronic HA is unable to replicate.

## Haemodynamic function

The modern human is the most encephalized of all species (Hadjistassou et al., 2015), allocating a disproportionate 20–25% of the body's total resting metabolic budget (Attwell et al., 2010) to maintain cerebral bioenergetic function. Given this high vulnerability to failure, it would appear evolution has favoured physiological 'defence' mechanisms capable of maintaining adequate $O_2$ and glucose delivery during times of oxygen deprivation (Bailey, 2019a).

In the present study we combined transcranial Doppler and duplex ultrasound to examine intracranial velocities

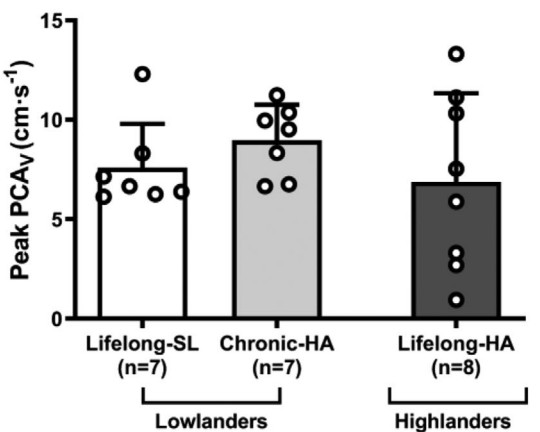
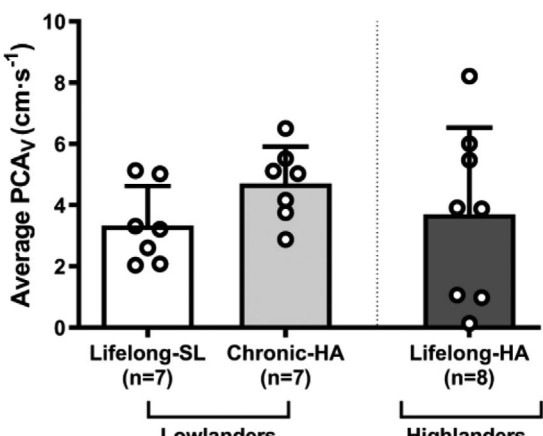
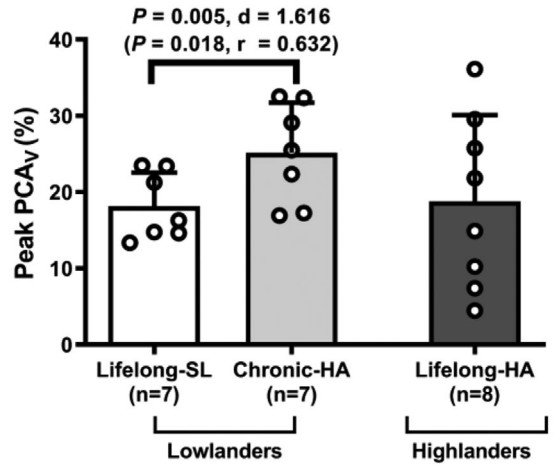
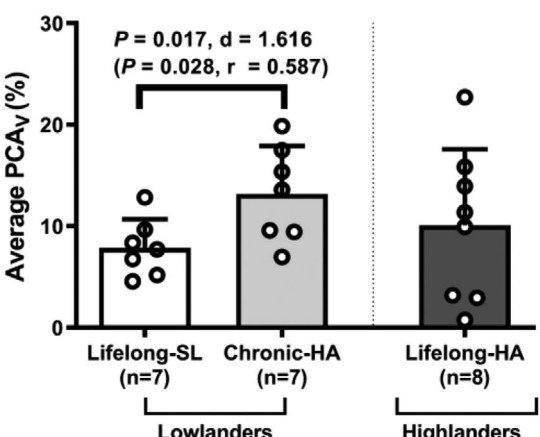

**Figure 6. Neurovascular coupling**
Values are means ± SD based on 9 lowlanders and 9 highlanders. HA, high altitude; PCA$_V$, posterior cerebral artery velocity; SL, sea level. Comparisons performed using a combination of parametric tests (paired samples *t*-tests for lowlander comparison and independent samples *t*-tests for highlander comparisons). Non-parametric equivalents are also provided in brackets (Wilcoxon signed rank and Mann-Whitney *U* tests). *d*, Cohen's (parametric) and *r*, Wilcoxon's (non-parametric) effect sizes.

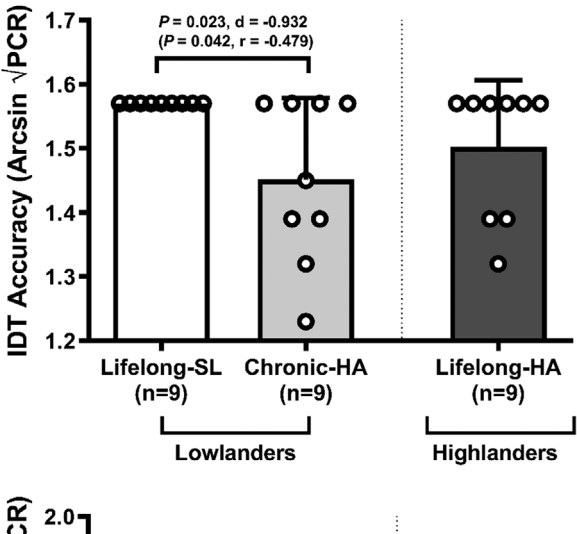

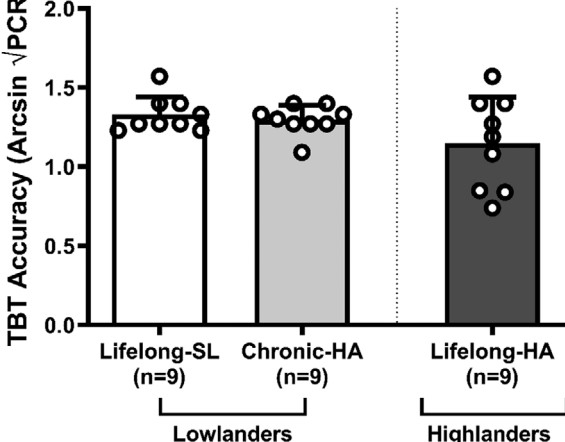

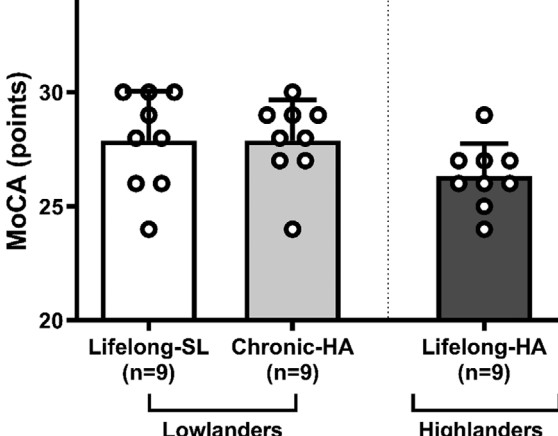

**Figure 7. Cognitive function**
Values are means ± SD based on 9 lowlanders and 9 highlanders. IDT, identification test; MoCA, Montreal Cognitive Assessment; SL, sea level; TBT, two-back test. Comparisons performed using a combination of parametric tests (paired samples *t*-tests for lowlander comparison and independent samples *t*-tests for highlander comparisons). Non-parametric equivalents also provided in brackets (Wilcoxon signed rank and Mann-Whitney *U* tests). *d*, Cohen's (parametric) and *r*, Wilcoxon's (non-parametric) effect sizes.

of the MCA and PCA and extracranial flow of the ICA and VA, respectively, to ascertain if regional differences in cerebral substrate delivery exist following acclimatization in lowlanders and lifelong HA in highlanders. We observed no differences in regional CBV or CBF between groups and the values obtained in lowlanders following chronic HA were comparable to those observed at SL. With the exception of marked reductions in $PCA_V$, these findings agree with previous research (Howe et al., 2019; Willie et al., 2014). In contrast, both MCA and PCA PI were lower in highlanders relative to lowlanders at SL and chronic HA. These differences may have contributed at least in part to the observed improvement in NVU integrity since exaggerated shear stress and pulsatility can disrupt endothelial tight junctions and increase BBB permeability (Garcia-Polite et al., 2017).

Measurements of basal CBF and substrate delivery were further complimented by TFA metrics of dCA to provide additional insight into region-specific pressure–flow coupling dynamics. A reduction in gain and reciprocal elevation in phase were collectively taken to reflect improved dCA, indicative of a less pressure-passive relationship between MAP and CBF ($MCA_V$/$PCA_V$) (Bailey et al., 2019). That we observed a reduction in PCA LF and VLF gain in lowlanders following chronic HA and in highlanders, provides further evidence for differential regulation of the posterior cerebral circulation in hypoxia (Ogoh et al., 2013; Subudhi et al., 2014; Willie et al., 2012), which may act to protect the phylogenetically older, arguably more important/sensitive regions of the brain (i.e. putamen, brain stem, thalamus, caudate nucleus, nucleus accumbens and pallidum).

Highlanders presented with a lower LF phase in the setting of lower (LF) gain (the latter reflecting smaller oscillations in MCAv for any given change in MAP) in both the anterior (MCA) and posterior (PCA) cerebral circulation compared to lowlanders at SL. Clearly, this differential response cannot simply be ascribed to the expected, more pronounced relative hypocapnia observed in highlanders, a stimulus known to improve pressure–flow coupling subsequent to changes in the respiratory chemoreflex (Ogoh et al., 2010) and/or increased sympathetic tone (Lundby et al., 2018).

The reduction in LF gain agrees with recent findings albeit selectively confined to the anterior circulation in Bolivian highlanders in response to 'driven' (squat-to-stand) oscillations (Bailey et al., 2019). That the highlander brain appears better equipped to buffer rapid surges and falls in arterial pressure suggests that this may reflect an additional neuroprotective adaptation. To what extent this was a cause or merely a consequence of preserved NVU integrity that was seen to be structurally tighter/more intact in the highlanders remains to be established, albeit beyond the scope of the present

study. Improved pressure–flow coupling has traditionally been interpreted to provide improved protection of the NVU and bioenergetic defense arising either as a result of hypoperfusion or hyperperfusion-mediated BBB disruption and consequent vasogenic oedema that can limit local $O_2$ diffusion (feedforward mechanism) known to incapacitate, or even prove fatal, to the less-adapted lowlander (Bailey, Bartsch et al., 2009). However, it is equally conceivable that preserved NVU structure and function may have contributed to the lower MCAv LF gain (feedback mechanism) observed given emerging roles for astrocytes and microglia as important intra-cranial baroreceptors/$O_2$ sensors that directly regulate CBF to optimize bioenergetics in response to altered neuronal activity, hypercapnia and hypoperfusion subsequent to autonomic sympathetic activation or P2Y12 receptor signalling (Csaszar et al., 2022; Marina et al., 2020).

In healthy humans, $CO_2$ is the most potent regulator of CBF, reflected by a ∼3–6% increase in blood flow per mmHg elevation in $CO_2$ (Sato et al., 2012; Skow et al., 2013; Willie et al., 2012). Regional differences in the magnitude of $CVR_{CO_2}$ are known to exist between the anterior and posterior circulation (Skow et al., 2013), but when scaled to resting blood flow and/or blood velocity and represented as a percentage change from baseline, this difference is eliminated (Hoiland et al., 2015; Sato et al., 2012; Willie et al., 2012). Moreover, a lower $CVR_{CO_2}$ has been consistently observed in patients with cognitive impairment (Beishon et al., 2017) and dementia (Cantin et al., 2011; den Abeelen et al., 2014; Kuwabara et al., 1992). In the present study, chronic HA in lowlanders increased MCA $CVR_{CO_2}$ and ICA $CVR_{CO_2}$ in agreement with prior studies (Fan et al., 2014; Fluck et al., 2015; Jensen et al., 1996), whereas posterior reactivity (VA $CVR_{CO_2}$ and PCA $CVR_{CO_2}$) remained unchanged. Furthermore, contrary to previous research in Andean (Bailey et al., 2019; Appenzeller et al., 2004) highlanders although consistent with MCA $CVR_{CO_2}$ assessed in Tibetan highlanders (Jansen et al., 1999), the present study also demonstrated elevated ICA $CVR_{CO_2}$ in highlanders when compared to lowlanders at SL. It is unclear to what extent these findings can be attributed to increased vascular NO bioavailability (Fan et al., 2019; Zimmermann & Haberl, 2003) albeit in the light of recent evidence suggesting that NO may not be obligatory for steady-state $CVR_{CO_2}$ (Hoiland et al., 2022) and/or hypoxia-induced sympathetic activation given that pharmacological blockade has been shown to decrease $CVR_{CO_2}$ (Przybyłowski et al., 2003). Notwithstanding the interpretive limitations of assuming linearity for the blood pressure response to hypercapnia (Hoiland, Fisher et al., 2019), ICA CVCi was also selectively elevated during the $CVR_{CO_2}$ challenge in the highlanders

implying a more 'reactive' pressure-induced elevation in (hypercapnia-induced) CBF which may potentially confer a selective advantage to lifelong hypoxia. Importantly, the heterogeneity observed in the $CVR_{CO_2}$ response(s) to hypoxia emphasizes the need to incorporate regional $CVR_{CO_2}$ assessments in future studies.

The NVU possesses an array of unique functional characteristics with the close interaction between the neuron and the astrocyte glial cell arguably its most pronounced feature (Schaeffer & Iadecola, 2021). Here, NVC controls local changes in functional brain activity to be tightly coupled with cerebral perfusion to maintain cerebral homeostasis. For the first time, we demonstrate that NVC remains preserved in highlanders despite the stress of lifelong HA hypoxia. These findings are consistent with previous research in lowlanders that concluded there was a negligible effect of hypoxia on the NVC response (Caldwell et al., 2017; Caldwell et al., 2018; Leacy et al., 2018; Lefferts et al., 2020), which may be attributable to the elevated concentrations of NO, given its pivotal role in the regulation of NVC (Hoiland et al., 2020).

## Cognitive function

A disruption to cerebral $O_2$/glucose homeostasis has the potential to impair cognition subsequent to reduced neurotransmitter release owing to reduced mono-amine synthesis, elevated glutamate excitotoxicity and alterations in choline acetyltranserase/acetyl cholinesterase expression (Freeman & Gibson, 1988). Further, both hypoxia and OXNOS are established risk factors implicated in the pathophysiology of dementia (Sun et al., 2006; Wojsiat et al., 2018). In the present study we demonstrate cognition to remain relatively intact in native highlanders (see 'Experimental limitations'), whereas an impairment was observed in lowlanders confirmed by the lower accuracy score during the IDT. These findings are supported by our previous work (Bailey et al., 2019).

## Experimental limitations

Several limitations warrant careful consideration to provide a more balanced perspective to the present findings. We acknowledge the interpretive limitations associated with the relatively small sample sizes and associated loss of statistical power, including caveats associated with type II and M errors (Gelman & Carlin, 2014), which for select variables, including MoCA scores, limited our ability to detect post hoc differences between acclimatized lowlanders and highlanders (they tended to be lower in the latter). Owing to logistical

constraints, we were also unable to match groups at baseline for educational status, which likely affected the cognitive outcomes, although the sample sizes were too small to allow for any meaningful covariate analyses. Despite rigorous medical screening and attainment of all inclusion criteria, two participants (one lowlander and one highlander) were diagnosed with borderline mild cognitive impairment (MCI) according to established criteria (Nasreddine et al., 2005), reflected by MoCA scores of 24–25 points. Despite this, we specifically chose to not exclude these participants given that they were apparently healthy albeit with (slightly lower) scores that fell within the boundaries of normal (biological) variation expected from this population for their given age and level of education (Rossetti et al., 2011). While we acknowledge the need to be more inclusive given that evolution to life at HA is clearly not selectively constrained to males alone, we specifically chose *a priori* not to focus on females given the additional logistical challenges posed by attempting to control for basal differences in circulating oestrogen, which is known to impact redox regulation of systemic and cerebrovascular function (Bailey, Culcasi et al., 2022). Equally, emerging evidence suggests that an inverse relationship may exist between circulating testosterone and oxidative-inflammatory stress (Maggio et al., 2006; Rovira-Llopis et al., 2017), notwithstanding the ability of testosterone to affect smooth muscle vasodilation directly (Webb et al., 1999) and upregulate aquaporin-4 expression on astrocytic endfeet (Gu et al., 2003) that could conceivably impact cerebrovascular function. However, it remains unclear to what extent basal differences in testosterone could have influenced the findings of the present study given the controversy reported in the literature in both lowlanders (Friedl et al., 1988; Vasankari et al., 1993) and native highlanders (Garmendia et al., 1982; Gonzales et al., 2009) exposed to HA. Furthermore, we were unable to (more) directly assess BBB integrity by employing dynamic contrast-enhanced and dynamic susceptibility contrast MRI (Tofts & Kermode, 1991) or via quantification of CSF to blood NVU proteins and/or transcerebral exchange (arterial-to-jugular venous concentration) gradients (Bailey, Bain et al., 2022; Lindblad et al., 2020), hence the focus on 'surrogate' albeit validated peripheral biomarkers (Janigro et al., 2020). Finally, it would have been of interest to assess cerebral tissue $O_2$ consumption/metabolism using near infra-red spectroscopy and/or derivation of cerebral mitochondrial $P_{O_2}$ inferred from arterial and jugular venous blood gas data, given prior research documenting that local reductions in (mitochondrial) $P_{O_2}$ may serve as the upstream (potentially hormetic) signal underlying systemic/cerebral free radical formation (Bailey et al., 2011; Bailey et al., 2018; Bailey et al., 2009; Woodside et al., 2014).

## Conclusions

Compared to lowlanders at SL, highlanders exhibited elevated basal plasma and red blood cell NO bioavailability, improved anterior and posterior dCA and elevated anterior $CVR_{CO2}$, preserved cerebral substrate delivery, NVC and cognition. In highlanders, S100B, NF-L and T-tau were consistently lower than and cognition comparable to lowlanders following chronic HA. Collectively, these findings highlight novel integrated adaptations towards the regulation of the NVU in highlanders that may represent a neuroprotective phenotype underpinning successful adaptation to the lifelong stress of HA hypoxia.

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

## Additional information

### Data availability statement

Original data arising from this research are available directly from D.M.B. upon reasonable request.

### Competing interests

D.M.B. is Editor-in-Chief of *Experimental Physiology*, Chair of the Life Sciences Working Group and member of the Human Spaceflight and Exploration Science Advisory Committee to the European Space Agency, Space Exploration Advisory Committee to the UK Space Agency, South East Wales Vascular Network, National Cardiovascular Network and Concussion in Sport Committee for The Department for Digital, Culture, Media & Sport (c/o Medical Research Council). D.M.B. is also affiliated to the companies FloTBI Inc., BrainEx Inc. and OrgEx Inc. focused on the technological development of novel biomarkers of brain injury in humans.

### Author contributions

Concept and design: B.S.S., R.L.H., M.M.T., F.C.V., P.N.A. and D.M.B.; acquisition, analysis and interpretation: all authors; drafting of the manuscript: B.S.S and D.M.B; critical revision of the manuscript for important intellectual content: all authors; statistical analysis: B.S.S. and D.M.B.; obtained funding:

F.C.V., P.N.A. and D.M.B; administrative, technical, or material support: B.S.S, R.L.H., M.M.T., F.C.V., P.N.A. and D.M.B.; supervision: F.C.V., P.N.A. and D.M.B. All authors have read and approved the final version of this manuscript and agree to be accountable for all aspects of the work in ensuring that questions related to the accuracy or integrity of any part of the work are appropriately investigated and resolved. All persons designated as authors qualify for authorship, and all those who qualify for authorship are listed.

## Funding

D.M.B. is supported by a Royal Society Wolfson Research Fellowship (WM170007) and funding from Higher Education Funding Council for Wales (to support B.S.S.). R.L.H. and P.N.A. are supported by NSERC and P.N.A. is a Canada Research Chair in Cerebrovascular Physiology. F.C.V. is supported by the Wellcome Trust (107544/Z/15/Z).

## Acknowledgements

The authors acknowledge the cheerful participation of participants.

## Keywords

acclimatization, cerebrovascular function, cognition, free radicals, high altitude, neurovascular unit

## Supporting information

Additional supporting information can be found online in the Supporting Information section at the end of the HTML view of the article. Supporting information files available:

**Statistical Summary Document**
**Peer Review History**

