## [Peer Review History · The Journal of Physiology]

Lifelong exposure to hypoxia in humans is associated with improved redox homeostasis and structural-functional adaptations of the neurovascular unit

Benjamin S Stacey, Ryan L Hoiland, Hannah G Caldwell, Connor A Howe, Tyler D Vermeulen, Michael M Tymko, Gustavo A Vizcardo-Galindo, Daniela Bermudez, Romulo Figueroa-Mujica, Chris Gasho, Eduoard Tuailon, Christophe Hirtz, Sylvain Lehmann, Nicola Marchi, Hayato Tsukamoto, Francisco C Villafuerte, Philip N Ainslie, and Damian Miles Bailey

DOI: 10.1113/JP283362

Corresponding author(s): Damian Bailey (damian.bailey@southwales.ac.uk)

Review Timeline:

Submission Date:	01-Jun-2022
Editorial Decision:	25-Jul-2022
Revision Received:	05-Oct-2022
Editorial Decision:	04-Nov-2022
Revision Received:	28-Nov-2022
Accepted:	20-Dec-2022

Senior Editor: Laura Bennet

Reviewing Editor: Ken O'Halloran

Transaction Report:

Dear Professor Bailey,

Re: JP-RP-2022-283362 "Functional-structural remodelling of the neurovascular unit in Andean highlanders" by Benjamin S Stacey, Ryan L Hoiland, Hannah G Caldwell, Connor A Howe, Tyler D Vermeulen, Michael M Tymko, Gustavo A Vizcardo-Galindo, Daniella Bermudez, Romulo Figueroa-Mujica, Chris Gasho, Eduoard Tuailon, Christophe Hirtz, Sylvain Lehmann, Nicola Marchi, Hayato Tsukamoto, Francisco C Villafuerte, Philip N Ainslie, and Damian Miles Bailey

Thank you for submitting your manuscript to The Journal of Physiology. It has been assessed by a Reviewing Editor and by 2 expert Referees and I am pleased to tell you that it is considered to be acceptable for publication following satisfactory revision.

The reports are copied at the end of this email. Please address all of the points and incorporate all requested revisions, or explain in your Response to Referees why a change has not been made.

NEW POLICY: In order to improve the transparency of its peer review process The Journal of Physiology publishes online as supporting information the peer review history of all articles accepted for publication. Readers will have access to decision letters, including all Editors' comments and referee reports, for each version of the manuscript and any author responses to peer review comments. Referees can decide whether or not they wish to be named on the peer review history document.

Authors are asked to use The Journal's premium BioRender (<https://biorender.com/>) account to create/redraw their Abstract Figures. Information on how to access The Journal's premium BioRender account is here: <https://physoc.onlinelibrary.wiley.com/journal/14697793/biorender-access> and authors are expected to use this service. This will enable Authors to download high-resolution versions of their figures. The link provided should only be used for the purposes of this submission. Authors will be charged for figures created on this premium BioRender account if they are not related to this manuscript submission.

I hope you will find the comments helpful and have no difficulty returning your revisions within 4 weeks.

Your revised manuscript should be submitted online using the links in Author Tasks Link Not Available.

Any image files uploaded with the previous version are retained on the system. Please ensure you replace or remove all files that have been revised.

REVISION CHECKLIST:

- Article file, including any tables and figure legends, must be in an editable format (eg Word)
- Abstract figure file (see above)
- Statistical Summary Document
- Upload each figure as a separate high quality file
- Upload a full Response to Referees, including a response to any Senior and Reviewing Editor Comments;
- Upload a copy of the manuscript with the changes highlighted.

- A potential 'Cover Art' file for consideration as the Issue's cover image;
- Appropriate Supporting Information (Video, audio or data set https://jp.msubmit.net/cgi-bin/main.plex?form_type=display_requirements#supp).

To create your 'Response to Referees' copy all the reports, including any comments from the Senior and Reviewing Editors, into a Word, or similar, file and respond to each point in colour or CAPITALS and upload this when you submit your revision.

I look forward to receiving your revised submission.

If you have any queries please reply to this email and staff will be happy to assist.

Yours sincerely,

Professor Laura Bennet
Senior Editor
The Journal of Physiology
<https://jp.msubmit.net>
<http://jp.physoc.org>
The Physiological Society
Hodgkin Huxley House
30 Farringdon Lane
London, EC1R 3AW
UK
<http://www.physoc.org>
<http://journals.physoc.org>

REQUIRED ITEMS:

-Author photo and profile. First (or joint first) authors are asked to provide a short biography (no more than 100 words for one author or 150 words in total for joint first authors) and a portrait photograph. These should be uploaded and clearly labelled with the revised version of the manuscript. See Information for Authors for further details.

-You must start the Methods section with a paragraph headed Ethical Approval. If experiments were conducted on humans confirmation that informed consent was obtained, preferably in writing, that the studies conformed to the standards set by the latest revision of the Declaration of Helsinki, and that the procedures were approved by a properly constituted ethics committee, which should be named, must be included in the article file. If the research study was registered (clause 35 of the Declaration of Helsinki) the registration database should be indicated, otherwise the lack of registration should be noted as an exception (e.g. The study conformed to the standards set by the Declaration of Helsinki, except for registration in a database.). For further information see: <https://physoc.onlinelibrary.wiley.com/hub/human-experiments>

-Please upload separate high-quality figure files via the submission form.

-Please ensure that any tables are in Word format and are, wherever possible, embedded in the article file itself.

-Please ensure that the Article File you upload is a Word file.

-A Statistical Summary Document, summarising the statistics presented in the manuscript, is required upon revision. It must be on the Journal's template, which can be downloaded from the link in the Statistical Summary Document section here: https://jp.msubmit.net/cgi-bin/main.plex?form_type=display_requirements#statistics

-Please include an Abstract Figure. The Abstract Figure is a piece of artwork designed to give readers an immediate understanding of the research and should summarise the main conclusions. If possible, the image should be easily 'readable' from left to right or top to bottom. It should show the physiological relevance of the manuscript so readers can assess the importance and content of its findings. Abstract Figures should not merely recapitulate other figures in the manuscript. Please try to keep the diagram as simple as possible and without superfluous information that may distract from the main conclusion(s). Abstract Figures must be provided by authors no later than the revised manuscript stage and should be uploaded as a separate file during online submission labelled as File Type 'Abstract Figure'. Please ensure that you include the figure legend in the main article file. All Abstract Figures should be created using BioRender. Authors should use The Journal's premium BioRender account to export high-resolution images. Details on how to use and access the premium account are included as part of this email.

EDITOR COMMENTS

Reviewing Editor:

Thank you for submitting your manuscript to The Journal of Physiology. Two experts in the field have reviewed the work. Constructive critiques are provided, which should be carefully addressed. Both referees draw focus to the use of peripheral blood markers of brain injury and potential limitations to this approach. I agree with referee 1 that the manuscript title should be reconsidered. Referee 2 draws focus to the central premise of the study, which should be clearly presented in the Introduction. The novelty of the present study should be clearly presented in the Discussion, Abstract, and Key Points sections.

In addition, on line 133 please include "..., except for registration in a database" or provide details if this study was a registered clinical trial. Concerning informed consent, can you please comment on written and verbal fluency in English in highlander participants (or perhaps all participants), or elaborate on whether documents were in the native language, or a translator employed to facilitate discussions. I appreciate that this would have been carefully considered by ethical boards but a comment in the manuscript would be useful.

Line 152 (and throughout): please refer to sex-matched not gender-matched.

Please include details of N, and statistical tests and exact p-values in legends (Tables and Figures).

Please provide a statistical summary document with your revised submission.

REFeree COMMENTS

Referee #1:

Stacey et al. examined differences in functional cerebral vascular physiology and regulatory factors between humans living at low and high altitude. Male participants were recruited and tested at low altitude (lowlanders) and high altitude (lowlanders and highlanders). Circulating factors contributing to cerebral blood flow control and neurovascular coupling were assessed at rest (free radicals, nitric oxide metabolites, markers of cerebral tissue injury), in addition to measurements of cerebral blood flow control (dynamic arterial pressure-cerebral blood flow relationships, CO₂ reactivity, and neurovascular coupling) and cognition. The authors report that compared with lowlanders at high altitude, highlanders exhibit elevated nitric oxide bioavailability, lower concentrations of injury biomarkers, improved cerebral blood flow control (buffering arterial pressure fluctuations, and responsiveness to both metabolically derived, and exogenous CO₂), and preserved cognition.

Examining cerebral vascular control mechanisms across humans at high and low altitude is an intriguing approach to understanding evolutionary adaptations to sustained hypoxia. This study is novel, and examines inter-connections of regulatory mechanisms controlling cerebral blood flow and delivery of metabolic substrates. The reported findings add to the growing body of literature exploring physiological differences between humans at high and low altitude. The authorship team are very well known in this field, and they have employed state of the art techniques to explore these research questions.

I have some comments for consideration for the authorship team:

1. The authors report that cognition was preserved in highlanders compared with lowlanders at low and high altitude. However, the Montreal Cognitive Assessment (MoCA) data in figure 5 show a clear reduction in highlanders (P=0.056). Please comment on this finding. Furthermore, please comment on the participants in both groups who exhibited MoCA scores below the "normal" threshold of 26.

2. Only males were recruited and tested in this study. Please provide the rationale for this approach, and the limitations of not being more inclusive when evolution to life at high altitude is not limited to males only. Indeed, it would be interesting to examine sex differences in these responses, particularly if hormonal differences influence cerebral regulatory control.
3. Please comment on the validity of measuring brain-specific markers from the peripheral venous blood. Do you have evidence that concentrations of these markers show similar responses when measured from the circulation draining the brain (e.g., IJV) vs. the periphery?
4. While measurements of oxygen delivery were examined, data assessing cerebral tissue oxygen consumption/metabolism are not included. Please provide your thoughts about this topic in relation to the reported findings, particularly as these assessments have been made in similar studies from this authorship team.
5. The term "structural remodelling" in the manuscript title is a little deceiving, as I was expecting markers of anatomical structure (such as with high resolution imaging), rather than circulating markers. Consider modifying the title to more accurately reflect the measurements made in this study.
6. Please report the age range, height, and weight of participants in each group
7. How were developmental delays and/or learning difficulties assessed in order to exclude participants?
8. Were any cardiovascular risk factors assessed, and if so, were any of these factors used as exclusion criteria (e.g., hypertension, diabetes, high cholesterol)?
9. Were any of the lowlander participants using carbonic anhydrase inhibitors at high altitude?
10. What volume of blood was withdrawn for assessment of circulating markers?
11. The terminology used to describe calculations of CVR and pulsatility are confusing; "sum" implies an addition (lines 257-258). CVR should simply be stated as MAP divided by CBV, and pulsatility should be the difference between systolic and diastolic velocities divided by mean velocity.
12. Add the term "artery" when defining ICA and VA. Also, please provide additional details about measurement of ICA and VA flow, including how velocity was analyzed; only mention of edge detection software is provided.
13. More details are required for the CVR to CO₂ description (lines 299-307). The end-tidal forcing technique should be introduced as the method of inducing controlled hypercapnia, with reference to the relevant publications on this method. What was the target etCO₂ during the hyperventilatory period? Also, if arterial pressure increased during the CO₂ reactivity tests, how was this accounted for in the subsequent analysis?
14. In the statistical analysis section (lines 327-331), outline the global questions that can be answered with each pair-wise comparison (i.e., why was it important to examine lowlander responses at both low and high altitude; why was it important to examine lowlanders at low altitude vs. highlanders at high altitude?).
15. In addition to reporting the transfer function phase and gain between arterial pressure and cerebral blood flow, please also report the power spectral density data for MAP, MCAv, and PCAv. This will provide an indicator of how oscillations in arterial pressure and cerebral blood flow changed from low to high altitude in the lowlanders, and whether there were differences between lowlanders and highlanders at high altitude.
16. In addition to oxygen content, also report oxygen delivery as this is discussed throughout the manuscript, and is easily calculated from the reported measurements.
17. Lower gain between arterial pressure and cerebral blood flow indicates better buffering of both increases and decreases in arterial pressure (line 430). Protecting the brain from hypoperfusion is just as important as hyperperfusion.
18. Include the number of participants for each measurement in each figure legend
19. Figure 3 contains 16 panels, which are small and difficult to read. Consider splitting this figure into 3 for dCA, CVR-CO₂ and NVC.
20. While a minor point that can be addressed in the copy-editing process, using paragraphs in the introduction and discussion would improve readability of these sections during the review process.

Referee #2:

This study, the authors examined a wide array of biomarkers, cognition, and cerebrovascular function in lowlanders at sea-level and following ascent to high altitude, and compared these changes to life-long highlanders. While I applaud the authors for such ambitious undertaking, the manuscript suffers from poor structure and the lack of a clear narrative. The introduction does not define the rationale of the study in sufficient detail. In addition, there are major issues surrounding the methodology and the quality of the data analysis perform.

Major comments

Introduction

The introduction in its current form does not clearly establish the premise of the study question. In the first paragraph, the authors should better describe the role of the neurovascular unit and its importance at high altitude.

The authors hypothesized that highlanders will exhibit evidence of adaptive NVU remodelling... improved cerebral substrate delivery and cerebrovascular function. As a single primary aim, there are too many seemingly disjointed array of parameters being examined. For example, how cerebrovascular CO₂ reactivity is relevant (presumably via the role of nitric oxide) has not been mentioned. Thus it seems out-of-place to be examined in this manuscript. The introduction, and ultimately manuscript as a whole, suffers from this. Many of the background have not been sufficiently established/described. This reviewer believes that this manuscript would benefit from having a more defined focus on neurovascular unit with chronic and life-long high altitude exposure. Alternatively, expand the introduction to better set the scene/scope of this study, and provide clear primary and secondary aims.

Methods

The authors stated that 'recruited nine age- and gender-matched healthy Andean (Aymara) participants (Table 1)'. Please include participant characteristics in Table 1 (age/gender/BMI).

The authors stated that " an ultrasensitive panel of blood-borne NVU proteins" was used. It is unclear how the authors can be certain that the changes in UCH-L1 and NFL are exclusively due to NVU damage, and not associated with neuronal damages outside of the CNS? (i.e., in the peripheral nervous system). Likewise, serum S100 β is used in this study as a biomarker of BBB permeability, yet there is evidence that cardiac glial release of S100 β (5). The studies cited in the Brain specific protein section include one review and one human study which reported no changes in these brain specific proteins following repeated micro- and hypergravity parabolic flights. It might be more useful to reference basic science studies which have validated proteins as highly specific biomarkers of neurovascular unit and BBB damage.

The authors stated that "Shapiro-Wilk W tests confirmed that all data sets were normally distributed." But looking at some of the figures (e.g., Figure 3, MCAv LF phase chronic high-altitude, MCA LF phase life-long high altitude, peak PCAv sea level; Figure 4 NFL, chronic high-altitude group etc), does not appear to be normally distributed. Sample size of 9 in each group is rather low and presents an unsolvable problem for parametric tests. Beyond low power, they prevent the assumption of normality, thus making the results with parametric tests unreliable. Furthermore, normality tests have low power to detect deviation from normality with small sample sizes. Please consider performing non-parametric tests and use appropriate corrections for multiple pairwise comparisons to reduce the risk of false positives.

What about outliers? For example Figure 1, HbNO for life-long high-altitude.

Please consider calculating and presenting effect sizes (4).

Discussion

My comments for the introduction also apply to the discussion. In its current form, the discussion reads rather disjointed and convoluted. It is difficult to discern which findings are novel and which are merely confirms previous observations. I recommend the authors focus on the novel data surrounding neurovascular unit in high altitude residents. Extensive revisions are required to improve clarity and flow.

There is some evidence that NVU play a role in limiting cerebral hyperperfusion during increases in blood pressure (3). Could the authors comment on whether a preserved NVU may account for the lower MCAv LF gain observed in highlanders? And whether such improved autoregulation may be responsible for the more intact BBB in highlanders? Indeed, a preserved NVU would provide a mechanistic explanation for lower MCAv LF gain, rather than the other way round as the authors are trying to imply.

In the second paragraph, the authors discuss the lower A- before mentioning a reciprocal increase in NO bioavailability. Do the authors think the lower A- at high altitude may be due to the role of NO as a reactive oxide species scavenger (1)? Previous studies have found increasing NO bioavailability increased cerebrovascular CO2 reactivity (2, 6). Could the increased NO bioavailability with high altitude account for the increased CO2 reactivity observed?

Minor comments

Methods

It is difficult to see the individual data points in the figures. Please change the display of the individual data points from aligned dot plot to scattered dot plot to help better visual the data points.

Discussion

It is often difficult to tell whether the authors are referring to findings in lowlanders following chronic high altitude or highlanders. Perhaps a better group designation would help avoid confusion.

References

1. Fago A, and Jensen FB. Hypoxia tolerance, nitric oxide, and nitrite: lessons from extreme animals. *Physiology (Bethesda)* 30: 116-126, 2015.
2. Fan JL, O'Donnell T, Gray CL, Croft K, Noakes AK, Koch H, and Tzeng YC. Dietary nitrate supplementation enhances cerebrovascular CO2 reactivity in a sex-specific manner. *J Appl Physiol (1985)* 127: 760-769, 2019.
3. Marina N, Christie IN, Korsak A, Doronin M, Brazhe A, Hosford PS, Wells JA, Sheikhabaehi S, Humoud I, Paton JFR, Lythgoe MF, Semyanov A, Kasparov S, and Gourine AV. Astrocytes monitor cerebral perfusion and control systemic circulation to maintain brain blood flow. *Nature communications* 11: 131, 2020.
4. Nakagawa S, and Cuthill IC. Effect size, confidence interval and statistical significance: a practical guide for biologists. *Biol Rev Camb Philos Soc* 82: 591-605, 2007.
5. Scherschel K, Hedenus K, Jungen C, Lemoine MD, Rubsamen N, Veldkamp MW, Klatt N, Lindner D, Westermann D, Casini S, Kuklik P, Eickholt C, Klocker N, Shivkumar K, Christ T, Zeller T, Willems S, and Meyer C. Cardiac glial cells release neurotrophic S100B upon catheter-based treatment of atrial fibrillation. *Sci Transl Med* 11: 2019.
6. Zimmermann C, and Haberl RL. L-arginine improves diminished cerebral CO2 reactivity in patients. *Stroke* 34: 643-647, 2003.

END OF COMMENTS

Confidential Review

01-Jun-2022

Response to Referees

We would like to take this opportunity to extend our sincere thanks to the Reviewing Editor and both referees for taking the time to review our manuscript. We agree with their comments/suggestions and have provided a brief response to each of the queries including modifications to the revised text (R1) highlighted in red.

Reviewing Editor:

Thank you for submitting your manuscript to The Journal of Physiology. Two experts in the field have reviewed the work. Constructive critiques are provided, which should be carefully addressed. Both referees draw focus to the use of peripheral blood markers of brain injury and potential limitations to this approach. I agree with referee 1 that the manuscript title should be reconsidered. Referee 2 draws focus to the central premise of the study, which should be clearly presented in the Introduction. The novelty of the present study should be clearly presented in the Discussion, Abstract, and Key Points sections.

Thank you for these general comments. We have since amended the manuscript as kindly requested (please see responses below). We have also included a newly formed Experimental Limitations paragraph to the Discussion to provide a more balanced perspective to our findings.

In addition, on line 133 please include "..., except for registration in a database" or provide details if this study was a registered clinical trial.

Thank you and this has been added.

Concerning informed consent, can you please comment on written and verbal fluency in English in highlander participants (or perhaps all participants), or elaborate on whether documents were in the native language, or a translator employed to facilitate discussions. I appreciate that this would have been carefully considered by ethical boards but a comment in the manuscript would be useful.

Our apologies for not having included this important information in the original submission. We have since clarified that all documents were translated into Spanish and participants spoke with a Peruvian research assistant to facilitate any discussions prior to obtaining written informed consent in Spanish.

Line 152 (and throughout): please refer to sex-matched not gender-matched. Our apologies and this has been amended throughout.

Please include details of N, and statistical tests and exact p-values in legends (Tables and Figures).

We have since clarified sample sizes and statistical tests employed in the legend of the Table and for all Figures. We have highlighted the P values and effect sizes in only those comparisons that were significant following application of parametric and non-parametric equivalents (the latter bracketed) in the Table and Figures, again, clarifying this within the legends. There is a great deal of information being conveyed here and we feel that this is the best approach to optimise readability, although we are open to any additional suggestions the referee(s) may have.

Please provide a statistical summary document with your revised submission. A statistical summary document has now been provided detailing all analyses performed. Please also note that we have expanded the Statistical Analysis section and as highlighted above, clarified all inferential analyses employed.

ADDITIONAL INFORMATION

Please note that we failed to include the following information in our original submission:

-*Cerebrovascular indices*: Cerebrovascular conductance, resistance and pulsatility indices (originally introduced in the Materials and Methods section). This information has now been added to the Results (text and Table 1) and Discussion sections including supporting statistical analyzes and references.

-*Neurovascular unit proteins*: Our apologies but we should have also included glial fibrillary acidic protein and total-tau data since these were also included in our original SIMOA platform analyzes. This information has now been added to the Results (text and Figure 2, two additional panels) and Discussion sections including supporting statistical analyzes and references. This additional information does not alter the qualitative outcomes of the study, rather, the findings, in particular the total-tau data, lend additional weight to our original interpretations/proposed mechanisms.

-*References*: Several additional references have been included in our repost to the referees.

REFeree COMMENTS

Referee #1:

Examining cerebral vascular control mechanisms across humans at high and low altitude is an intriguing approach to understanding evolutionary adaptations to sustained hypoxia. This study is novel, and examines inter-connections of regulatory mechanisms controlling cerebral blood flow and delivery of metabolic substrates. The reported findings add to the growing body of literature exploring physiological differences between humans at high and low altitude. The authorship team are very well known in this field, and they have employed state of the art techniques to explore these research questions. I have some comments for consideration for the authorship team.

Thank you for your constructive feedback and we appreciate your time and efforts reviewing our manuscript. We agree with your helpful comments/suggestions and have provided a brief response to each of the queries including modifications to the revised text (R1) highlighted in red.

The authors report that cognition was preserved in highlanders compared with lowlanders at low and high altitude. However, the Montreal Cognitive Assessment (MoCA) data in figure 5 show a clear reduction in highlanders ($P=0.056$). Please comment on this finding. Furthermore, please comment on the participants in both groups who exhibited MoCA scores below the "normal" threshold of 26.

Thank you for highlighting these two important points. Regarding the first point, we originally chose to not make too much of the lower MoCA score in the highlanders simply because this did not achieve significance. However, upon reflection and as rightfully stated, this is 'borderline' and certainly worthy of comment. Retrospective analyses revealed that the lack of significance compared to lowlanders chronic-HA was the likely consequence of inadequate statistical power (parametric: $P = 0.056$, $d = 0.973$, $1-\beta = 0.494$ and non-parametric: $P = 0.113$, $r = 0.391$). Closer inspection of the MoCA questionnaire also revealed that the highlanders were characterised by a lower educational status compared to their lowlander counterparts (13 ± 3 years vs. 20 ± 2 years, $P = <0.001$, $d = 3.119$), that may have contributed to the observed differences (we were not in a position to prospectively match for educational status, only age, sex and BMI). Regarding the second point, closer inspection of (newly revised) Figure 7 reveals that one lowlander (both at Sea-Level and Chronic High-Altitude) and one highlander presented with a MoCA score of 24-25 points, indicative of mild cognitive decline according to established guidelines (Nasreddine *et al.*, 2005). These important points have been added to the Results and Discussion sections, including a newly formed Experimental Limitations paragraph.

Only males were recruited and tested in this study. Please provide the rationale for this approach, and the limitations of not being more inclusive when evolution to life at high altitude is not limited to males only. Indeed, it would be interesting to examine sex differences in these responses, particularly if hormonal differences influence cerebral regulatory control.

Thank you for this comment and we agree that it would have been interesting to examine to what extent sex differences potentially impact our findings, that constitutes an entirely separate, albeit clinically relevant investigation. We specifically chose to constrain our focus exclusively to males due primarily to our inability to control for differences in circulating oestrogen known to impact redox-regulation of systemic and cerebrovascular function in

females (Bailey *et al.*, 2022b). While this was our intent *a priori*, we have added this information to the Experimental Limitations paragraph.

Please comment on the validity of measuring brain-specific markers from the peripheral venous blood. Do you have evidence that concentrations of these markers show similar responses when measured from the circulation draining the brain (e.g., IJV) vs. the periphery?

We were not in a position to perform transcerebral exchange (i.e. arterial to jugular venous concentration gradients) of these NVU-specific proteins owing to ethical constraints. However, our prior research albeit in a separate group of profoundly hypoxaemic-hypercapnic participants identified that an increase in the net cerebral output (i.e. more pronounced elevation in jugular venous outflow) scales with an increase in the systemic concentrations of an identical panel of NVU proteins (Bailey *et al.*, 2022a). Furthermore, we highlighted in the original text that for example, an elevation in the systemic concentration of S100B correlates directly with the extent and temporal sequence of BBB opening confirmed by contrast-enhanced MRI (Kanner *et al.*, 2003). However, there are nonetheless interpretive limitations associated with a reliance on peripheral biomarkers measured distal to the organ of primary interest (i.e. brain), including potential extraneous sources (a point also raised by the 2nd referee). This information including supporting references and suggestions for future research have been added to the Experimental limitations paragraph.

While measurements of oxygen delivery were examined, data assessing cerebral tissue oxygen consumption/metabolism are not included. Please provide your thoughts about this topic in relation to the reported findings, particularly as these assessments have been made in similar studies from this authorship team.

Thank you for this comment and our prior research has indeed focused on the reduction in (calculated) cerebral mitochondrial PO₂ (inferred from arterial and jugular venous blood gas data) as an 'upstream' hormetic catalyst underlying cerebral free radical formation: evidence for human cerebral (redox) O₂-sensing (Bailey *et al.*, 2006; Bailey *et al.*, 2018). As highlighted prior, we weren't in a position to perform these calculations in the present study given that our blood gas data were constrained solely to the (cephalic) venous circulation. We were also not in a position to collect near infra-red spectroscopic data to infer changes in cortical (de)oxygenation, although likewise, we have previously documented an inverse relationship between deoxyhaemoglobin signal intensity and systemic free radical formation (Woodside *et al.*, 2014). This information, including the potential mechanistic links and supporting references, have been added to the Experimental limitations paragraph.

The term "structural remodelling" in the manuscript title is a little deceiving, as I was expecting markers of anatomical structure (such as with high resolution imaging), rather than circulating markers. Consider modifying the title to more accurately reflect the measurements made in this study.

Thank you for this comment and our interpretation has now been altered throughout the manuscript including a title change to 'Lifelong exposure to high-altitude hypoxia in humans is associated with improved redox homeostasis and structural-functional adaptations of the neurovascular unit' to better reflect our primary findings.

Please report the age range, height, and weight of participants in each group

Thank you for these suggestions. This information (including supporting statistical analyses) has now been included in Table 1.

How were developmental delays and/or learning difficulties assessed in order to exclude participants?

Medical history was obtained during a thorough medical examination. This has been clarified in Materials and Methods, under the 'Participants' subheading.

Were any cardiovascular risk factors assessed, and if so, were any of these factors used as exclusion criteria (e.g., hypertension, diabetes, high cholesterol)?

Yes and again, our apologies for not having provided this in the original submission. This has been clarified in Materials and Methods, under the 'Participants' subheading. We have also taken the liberty of further clarifying that none of the participants were prescribed medication.

Were any of the lowlander participants using carbonic anhydrase inhibitors at high altitude?

Thank you for this question and no, all lowlanders abstained from any prophylactic medication at HA given that this interferes with natural acclimatisation, in particular the redox-regulation of systemic and cerebrovascular function (Bailey *et al.*, 2012). This has been clarified in Materials and Methods, under the 'Participants' subheading and a reference included.

What volume of blood was withdrawn for assessment of circulating markers?

A total of ~25 mL whole venous blood was withdrawn for assessment of circulating biomarkers. This has been clarified in Materials and Methods, under the 'Blood sampling' subheading.

The terminology used to describe calculations of CVR and pulsatility are confusing; "sum" implies an addition (lines 257-258). CVR should simply be stated as MAP divided by CBV, and pulsatility should be the difference between systolic and diastolic velocities divided by mean velocity.

Thank you for this suggestion. This has been clarified in Materials and Methods, in the Cerebrovascular function section, under the newly added 'Perfusion' subheading. Please also note that we failed to include the following information in our original submission:

-*Cerebrovascular indices*: Cerebrovascular conductance, resistance and pulsatility indices (originally introduced in the Materials and Methods section). This information has now been added to the Results (text and Table 1) and Discussion sections including supporting statistical analyses and references. Our apologies for this omission.

Add the term "artery" when defining ICA and VA. Also, please provide additional details about measurement of ICA and VA flow, including how velocity was analyzed; only mention of edge detection software is provided.

This section has been expanded extensively to provide critical information, including supporting references in the Cerebrovascular function, (modified) 'Perfusion' subheading.

More details are required for the CVR to CO₂ description (lines 299-307). The end-tidal forcing technique should be introduced as the method of inducing controlled hypercapnia,

with reference to the relevant publications on this method. What was the target etCO₂ during the hyperventilatory period? Also, if arterial pressure increased during the CO₂ reactivity tests, how was this accounted for in the subsequent analysis?

Thank you for this point and this section has now been extended to include the relevant information including supporting references. The section has been amended to include information relating to the dynamic end-tidal forcing system. The target PET_{CO₂} was +9 mmHg above the participants' baseline. As anticipated, mean arterial blood pressure did indeed increase during the CVR_{CO₂} challenge, however this was comparable across both conditions and groups (i.e. $P = > 0.050$). This information has been clarified in the Materials and Methods and Results sections under the CVR_{CO₂} subheadings.

In the statistical analysis section (lines 327-331), outline the global questions that can be answered with each pair-wise comparison (i.e., why was it important to examine lowlander responses at both low and high altitude; why was it important to examine lowlanders at low altitude vs. highlanders at high altitude?).

Thank you for this suggestion and this section has now been expanded to include this information.

In addition to reporting the transfer function phase and gain between arterial pressure and cerebral blood flow, please also report the power spectral density data for MAP, MCAv, and PCAv. This will provide an indicator of how oscillations in arterial pressure and cerebral blood flow changed from low to high altitude in the lowlanders, and whether there were differences between lowlanders and highlanders at high altitude.

Thank you for this comment. The corresponding PSDs (VLF and LF) for MAP, MCAv and PCAv have now been added to Figure 4, including the within/between group statistical comparisons (parametric + non-parametric).

In addition to oxygen content, also report oxygen delivery as this is discussed throughout the manuscript, and is easily calculated from the reported measurements.

Our apologies for this omission. Global cerebral oxygen delivery (gCDO₂) has now been included in Figure 3.

Lower gain between arterial pressure and cerebral blood flow indicates better buffering of both increases and decreases in arterial pressure (line 430). Protecting the brain from hypoperfusion is just as important as hyperperfusion.

Thank you for this comment and the sentence has now been amended to include this information. Please also note that we have provided further amendments to this section to comply with comments raised by the 2nd referee.

Include the number of participants for each measurement in each figure legend.

This information has been added.

Figure 3 contains 16 panels, which are small and difficult to read. Consider splitting this figure into 3 for dCA, CVR-CO₂ and NVC.

Thank you for this suggestion. Separate figures for dCA, CVR_{CO₂} and NVC have now been included as Figure 4, 5 and 6 (7 figures in total).

While a minor point that can be addressed in the copy-editing process, using paragraphs in the introduction and discussion would improve readability of these sections during the review process.

We have now included more paragraphs to these sections as kindly requested. Indeed, we have restructured both sections (with demarcating subheadings in the Discussion) to improve narrative structure and flow.

Formatted: Position: Horizontal: Center, Relative to: Margin, Vertical: 0", Relative to: Paragraph, Wrap Around

Referee #2:

This study, the authors examined a wide array of biomarkers, cognition, and cerebrovascular function in lowlanders at sea-level and following ascent to high altitude, and compared these changes to life-long highlanders. While I applaud the authors for such ambitious undertaking, the manuscript suffers from poor structure and the lack of a clear narrative. The introduction does not define the rationale of the study in sufficient detail. In addition, there are major issues surrounding the methodology and the quality of the data analysis performed.

Thank you for your constructive feedback and we appreciate your time and efforts reviewing our manuscript. We agree with your helpful comments/suggestions and have provided a brief response to each of the queries including modifications to the revised text (R1) highlighted in red. We have made a concerted effort to improve the structure that includes a clearer upfront definition of the rationale and underpinning methodology. We have also complimented our original parametric analyzes with non-parametric equivalents. Finally, we have included a newly formed Experimental Limitations paragraph to the Discussion (which has also been restructured extensively, including subheadings) to provide a more balanced perspective to our findings and improve narrative flow. Please see more specific responses below.

Introduction. The introduction in its current form does not clearly establish the premise of the study question. In the first paragraph, the authors should better describe the role of the neurovascular unit and its importance at high altitude.

The opening paragraph of the introduction has now been extended to highlight the importance of maintained redox homeostasis and (effective) NVU structure-function (including better description of constituent components) during the course of acclimatisation to the hypoxia of high-altitude (HA). This has been supported by relevant references.

Introduction. The authors hypothesized that highlanders will exhibit evidence of adaptive NVU remodelling... improved cerebral substrate delivery and cerebrovascular function. As a single primary aim, there are too many seemingly disjointed array of parameters being examined. For example, how cerebrovascular CO₂ reactivity is relevant (presumably via the role of nitric oxide) has not been mentioned. Thus it seems out-of-place to be examined in this manuscript. The introduction, and ultimately manuscript as a whole, suffers from this. Many of the background have not been sufficiently established/described. This reviewer believes that this manuscript would benefit from having a more defined focus on neurovascular unit with chronic and life-long high altitude exposure. Alternatively, expand the introduction to better set the scene/scope of this study, and provide clear primary and secondary aims.

We have now provided more backdrop to the redox-regulation of NVU structure-function highlighting the regulatory roles played by free radicals and associated reactive oxygen/nitrogen species in the preservation of neuronal, glial and vascular homeostasis. The final paragraph has been recrafted to provide more structure and context to the integrated components underlying NVU structure-function and we have better defined our primary aim and working hypotheses.

Methods. The authors stated that 'recruited nine age- and gender-matched healthy Andean

(Aymara) participants (Table 1)'. Please include participant characteristics in Table 1 (age/gender/BMI).

Participant demographics have been added to Table 1 including the statistical outcomes (parametric and non-parametric).

Methods. The authors stated that "an ultrasensitive panel of blood-borne NVU proteins" was used. It is unclear how the authors can be certain that the changes in UCH-L1 and NFL are exclusively due to NVU damage, and not associated with neuronal damages outside of the CNS? (i.e., in the peripheral nervous system). Likewise, serum S100 β is used in this study as a biomarker of BBB permeability, yet there is evidence that cardiac glial release of S100 β (5). The studies cited in the Brain specific protein section include one review and one human study which reported no changes in these brain specific proteins following repeated micro- and hypergravity parabolic flights. It might be more useful to reference basic science studies which have validated proteins as highly specific biomarkers of neurovascular unit and BBB damage.

Thank you for this important comment and we were originally referring to the analytical platforms deployed that, especially in the case of the single molecule array assay (Simoa), provides unprecedented detection sensitivity-specificity of the NVU proteins assessed herein. To that end, we have rewritten the 'NVU proteins' section in Materials and Methods with (more appropriate) supporting references. We have also expanded the Discussion section (please note that the NVU proteins paragraph(s) have been translocated to the newly formed Molecular function subheading) to highlight potential contributions from extracranial sources, including adipose tissue and cardiac glia for S100B, neuromuscular junction for UCH-L1 and erythrocytes for NSE, with supporting references. Furthermore, we have included a sentence in the Experimental limitations paragraph highlighting that we were unable to (more) directly assess BBB integrity by employing dynamic contrast-enhanced and dynamic susceptibility contrast MRI (Tofts & Kermode, 1991) or via quantification of CSF to blood NVU protein and/or transcerebral exchange (arterial-to-jugular venous concentration) gradients (Lindblad *et al.*, 2020; Bailey *et al.*, 2022a), hence the focus on 'surrogate' albeit validated peripheral biomarkers (Janigro *et al.*, 2021). We have also tempered our take-homes to reflect these interpretive constraints and to provide a more balanced narrative.

Methods. The authors stated that "Shapiro-Wilk W tests confirmed that all data sets were normally distributed." But looking at some of the figures (e.g., Figure 3, MCAV LF phase chronic high-altitude, MCA LF phase life-long high altitude, peak PCAV sea level; Figure 4 NFL, chronic high-altitude group etc), does not appear to be normally distributed. Sample size of 9 in each group is rather low and presents an unsolvable problem for parametric tests. Beyond low power, they prevent the assumption of normality, thus making the results with parametric tests unreliable. Furthermore, normality tests have low power to detect deviation from normality with small sample sizes. Please consider performing non-parametric tests and use appropriate corrections for multiple pairwise comparisons to reduce the risk of false positives. What about outliers? For example Figure 1, HbNO for life-long high-altitude. Please consider calculating and presenting effect sizes (4).

Thank you for these comments and we acknowledge the limitations associated with small sample sizes that we have highlighted in the Experimental limitations paragraph. We have complemented our parametric analyses with non-parametric equivalents (Wilcoxon signed rank and Mann-Whitney *U* tests for within and between group comparisons respectively) to

provide additional re-assurance of the differences observed and we have also provided corresponding effect sizes (Cohen's d and Wilcoxon r values). This information has been added to the Table/Figures and clarified within the Statistical analysis section and corresponding legends also. Please note that following (non-parametric) analysis, all primary findings remained, with the exception of S100B and A⁺. Non-parametric testing indicated no difference in S100B between lowlanders-(lifelong) SL and highlanders-lifelong-HA ($P = 0.094$, $r = -0.414$). However, when compared to lowlanders at chronic-HA, highlanders still exhibited lower S100B ($P = 0.040$, $r = -0.484$). Equally, the reduction observed in basal A⁺ between lowlanders at SL and lowlanders following chronic-HA also persisted, albeit borderline ($P = 0.051$, $r = -0.461$), the likely consequence of inadequate (statistical) power. We have also amended the Results and Discussion accordingly to reflect these subtle qualitative differences that did not influence the primary outcomes of this study. With regards the SNO-Hb 'outlier' exhibited by one of our highlanders, we believe this concentration to be correct and physiological (i.e. not an analytical anomaly) given the equivalent elevation in plasma concentrations of bioactive NO metabolites observed. Furthermore, removal of this single data point does not impact the qualitative outcome(s).

Discussion. My comments for the introduction also apply to the discussion. In its current form, the discussion reads rather disjointed and convoluted. It is difficult to discern which findings are novel and which are merely confirms previous observations. I recommend the authors focus on the novel data surrounding neurovascular unit in high altitude residents. Extensive revisions are required to improve clarity and flow.

Thank you for these helpful comments. We have reconfigured the Discussion providing subheadings highlighting each of the integrated foci. We have also taken the opportunity to re-order the Results section also (i.e. molecular to haemodynamic to clinical function, in that specific order, to align with the proposed mechanisms). Please note we have translocated the NVU integrity paragraph to the Molecular function section, following on from the OXNOS narrative given that this makes (better) intuitive sense and improves narrative flow. We have also sought to highlight how our work has extended the existing literature base, notably, that the novel integrated translational approach employed by the present study is the first to provide evidence for 'integrated NVU adaptation' in highlanders.

Discussion. There is some evidence that the NVU plays a role in limiting cerebral hyperperfusion during increases in blood pressure (3). Could the authors comment on whether a preserved NVU may account for the lower MCAv LF gain observed in highlanders? And whether such improved autoregulation may be responsible for the more intact BBB in highlanders? Indeed, a preserved NVU would provide a mechanistic explanation for lower MCAv LF gain, rather than the other way round as the authors are trying to imply.

Thank you for these insightful comments. Your feedforward/feedback suggestions are entirely conceivable given the paradigm shift in our understanding of the NVU 'construct' over the last two decades, evolving from a unidimensional process constrained to local neuronal-astrocytic signaling to a multidimensional one in which mediators released from multiple cells engage distinct signaling pathways and effector systems across the entire cerebrovascular network (Iadecola, 2017). Our original submission clearly failed to mechanistically 'link' aspects of improved dCA (i.e. lower LF Gain) to a structural tighter/more intact NVU (feedforward effect). This has now been rectified in the revised m/s and we have provided a more balanced interpretation of our (TFA) findings, including a brief explanation

as to why this reflects a neuroprotective adaptation within the context of protecting against 'both' hyper and hypoperfusion. Equally, we have briefly discussed the possibility that a preserved NVU may account for the lower MCAv LF Gain (feedback effect) highlighting the vasoregulatory roles and underlying mechanisms identified for astrocytes and microglia in the defense of cerebral perfusion and bioenergetics (Marina *et al.*, 2020; Csaszar *et al.*, 2022).

Discussion. In the second paragraph, the authors discuss the lower A- before mentioning a reciprocal increase in NO bioavailability. Do the authors think the lower A- at high altitude may be due to the role of NO as a reactive oxide species scavenger (1)?

Thank you for this comment. While thermodynamically plausible (Bailey *et al.*, 2009), the possibility that the lower A^{•-} observed following chronic hypoxia in lowlanders was related to enhanced NO scavenging (Singh *et al.*, 2012) is unlikely given that the elevated basal A^{•-} in highlanders persisted in the face of an equivalent elevation in vascular NO bioavailability. This information including a more specific reference (sodium nitrite supplementation in drinking water in a rodent model attenuates hypoxia-induced oxidative stress and modulates HIF-1a stability) (Singh *et al.*, 2012) has been clarified in the revised m/s.

Previous studies have found increasing NO bioavailability increased cerebrovascular CO₂ reactivity (2, 6). Could the increased NO bioavailability with high altitude account for the increased CO₂ reactivity observed?

This is indeed conceivable, although the literature is somewhat divided, with (our) recent evidence suggesting that NO is not necessarily obligatory for steady-state CVR_{CO₂} (Hoiland *et al.*, 2022). We have provided a more balanced interpretation of our findings by highlighting that it is unclear to what extent these findings can be attributed to increased vascular NO bioavailability (Zimmermann & Haberl, 2003; Fan *et al.*, 2019) albeit in the light of recent evidence suggesting that NO may not be obligatory for steady-state CVR_{CO₂} (Hoiland *et al.*, 2022) and/or hypoxia-induced sympathetic activation given that pharmacological blockade has been shown to decrease CVR_{CO₂} (Przybyłowski *et al.*, 2003).

Discussion. It is often difficult to tell whether the authors are referring to findings in lowlanders following chronic high altitude or highlanders. Perhaps a better group designation would help avoid confusion.

Thank you for bringing this to our attention and we agree. The groups have now been changed to the following differentiating exposure time from location: For lowlanders: Lifelong-Sea-Level (SL) and Chronic High-Altitude (HA); For highlanders: Lifelong-HA. The narrative, text and figures have been changed throughout to clarify these differentiations.

Figures. It is difficult to see the individual data points in the figures. Please change the display of the individual data points from aligned dot plot to scattered dot plot to help better visualise the data points.

Thank you for this suggestion. All figures have now been amended to illustrate scattered dot plots to help better visualise the data points.

Thank you for your additional references and please note we have also included a number of additional references (please see below).

Author's additional supporting references:

- Bailey DM, Bain AR, Hoiland RL, Barak OF, Drvis I, Hirtz C, Lehmann S, Marchi N, Janigro D, MacLeod DB, Ainslie PN & Dujic Z. (2022a). Hypoxemia increases blood-brain barrier permeability during extreme apnea in humans. *Journal of cerebral blood flow and metabolism : official journal of the International Society of Cerebral Blood Flow and Metabolism* **42**, 1120-1135.
- Bailey DM, Brugniaux JV, Pietri S, Culcasi M & Swenson ER. (2012). Redox regulation of neurovascular function by acetazolamide: complementary insight into mechanisms underlying high-altitude acclimatisation. *The Journal of physiology* **590**, 3627-3628.
- Bailey DM, Culcasi M, Filipponi T, Brugniaux JV, Stacey BS, Marley CJ, Soria R, Rimoldi SF, Cerny D, Rexhaj E, Pratali L, Salmon CS, Jauregui CM, Villena M, Villafuerte F, Rockenbauer A, Pietri S, Scherrer U & Sartori C. (2022b). EPR spectroscopic evidence of iron-catalysed free radical formation in chronic mountain sickness: Dietary causes and vascular consequences. *Free radical biology & medicine* **184**, 99-113.
- Bailey DM, Rasmussen P, Evans KA, Bohm AM, Zaar M, Nielsen HB, Brassard P, Nordborg NB, Homann PH, Raven PB, McEneny J, Young IS, McCord JM & Secher NH. (2018). Hypoxia compounds exercise-induced free radical formation in humans; partitioning contributions from the cerebral and femoral circulation. *Free Radical Biology & Medicine* **124**, 104-113.
- Bailey DM, Roukens R, Knauth M, Kallenberg K, Christ S, Mohr A, Genius J, Storch-Hagenlocher B, Meisel F, McEneny J, Young IS, Steiner T, Hess K & Bartsch P. (2006). Free radical-mediated damage to barrier function is not associated with altered brain morphology in high-altitude headache. *Journal of Cerebral Blood Flow and Metabolism* **26**, 99-111.
- Bailey DM, Taudorf S, Berg RM, Lundby C, McEneny J, Young IS, Evans KA, James PE, Shore A, Hullin DA, McCord JM, Pedersen BK & Moller K. (2009). Increased cerebral output of free radicals during hypoxia: implications for acute mountain sickness? *American journal of physiology Regulatory, integrative and comparative physiology* **297**, R1283-1292.
- Csaszar E, Lenart N, Cserep C, Kornyei Z, Fekete R, Posfai B, Balazsfi D, Hangya B, Schwarcz AD, Szabadits E, Szollosi D, Szigeti K, Mathe D, West BL, Sviatko K, Bras AR, Mariani JC, Kliewer A, Lenkei Z, Hricisak L, Benyo Z, Baranyi M, Sperlagh B, Menyhart A, Farkas E & Denes A. (2022). Microglia modulate blood flow, neurovascular coupling, and hypoperfusion via purinergic actions. *J Exp Med* **219**.
- Fan JL, O'Donnell T, Gray CL, Croft K, Noakes AK, Koch H & Tzeng YC. (2019). Dietary nitrate supplementation enhances cerebrovascular CO₂ reactivity in a sex-specific manner. *Journal of applied physiology* **127**, 760-769.

- Hoiland RL, Caldwell HG, Carr J, Howe CA, Stacey BS, Dawkins T, Wakeham DJ, Tremblay JC, Tymko MM, Patrician A, Smith KJ, Sekhon MS, MacLeod DB, Green DJ, Bailey DM & Ainslie PN. (2022). Nitric oxide contributes to cerebrovascular shear-mediated dilatation but not steady-state cerebrovascular reactivity to carbon dioxide. *J Physiol* **600**, 1385-1403.
- Iadecola C. (2017). The Neurovascular Unit Coming of Age: A Journey through Neurovascular Coupling in Health and Disease. *Neuron* **96**, 17-42.
- Janigro D, Bailey DM, Lehmann S, Badaut J, O'Flynn R, Hirtz C & Marchi N. (2021). Peripheral blood and salivary biomarkers of blood-brain barrier permeability and neuronal damage: clinical and applied concepts. *Frontiers in Neurology*.
- Kanner AA, Marchi N, Fazio V, Mayberg MR, Koltz MT, Siomin V, Stevens GH, Masaryk T, Aumayr B, Ayumar B, Vogelbaum MA, Barnett GH & Janigro D. (2003). Serum S100B: a noninvasive marker of blood-brain barrier function and brain lesions. *Cancer* **97**, 2806-2813.
- Lindblad C, Nelson DW, Zeiler FA, Ercole A, Ghatan PH, von Horn H, Risling M, Svensson M, Agoston DV, Bellander BM & Thelin EP. (2020). Influence of Blood-Brain Barrier Integrity on Brain Protein Biomarker Clearance in Severe Traumatic Brain Injury: A Longitudinal Prospective Study. *Journal of neurotrauma* **37**, 1381-1391.
- Marina N, Christie IN, Korsak A, Doronin M, Brazhe A, Hosford PS, Wells JA, Sheikhabaei S, Humoud I, Paton JFR, Lythgoe MF, Semyanov A, Kasparov S & Gourine AV. (2020). Astrocytes monitor cerebral perfusion and control systemic circulation to maintain brain blood flow. *Nat Commun* **11**, 131.
- Nasreddine ZS, Phillips NA, Bedirian V, Charbonneau S, Whitehead V, Collin I, Cummings JL & Chertkow H. (2005). The Montreal Cognitive Assessment, MoCA: a brief screening tool for mild cognitive impairment. *Journal of the American Geriatric Society* **53**, 695-699.
- Singh M, Arya A, Kumar R, Bhargava K & Sethy NK. (2012). Dietary nitrite attenuates oxidative stress and activates antioxidant genes in rat heart during hypobaric hypoxia. *Nitric Oxide* **26**, 61-73.
- Tofts PS & Kermode AG. (1991). Measurement of the blood-brain barrier permeability and leakage space using dynamic MR imaging. 1. Fundamental concepts. *Magn Reson Med* **17**, 357-367.
- Woodside JD, Gutowski M, Fall L, James PE, McEneny J, Young IS, Ogoh S & Bailey DM. (2014). Systemic oxidative-nitrosative-inflammatory stress during acute exercise in hypoxia; implications for microvascular oxygenation and aerobic capacity. *Experimental physiology* **99**, 1648-1662.

Zimmermann C & Haberl RL. (2003). L-arginine improves diminished cerebral CO₂ reactivity in patients. *Stroke* **34**, 643-647.

Formatted: Position: Horizontal: Center, Relative to: Margin, Vertical: 0", Relative to: Paragraph, Wrap Around

Dear Professor Bailey,

Re: JP-RP-2022-283362R1 "Lifelong exposure to hypoxia in humans is associated with improved redox homeostasis and structural-functional adaptations of the neurovascular unit" by Benjamin S Stacey, Ryan L Hoiland, Hannah G Caldwell, Connor A Howe, Tyler D Vermeulen, Michael M Tymko, Gustavo A Vizcardo-Galindo, Daniela Bermudez, Romulo Figueroa-Mujica, Chris Gasho, Eduoard Tuailon, Christophe Hirtz, Sylvain Lehmann, Nicola Marchi, Hayato Tsukamoto, Francisco C Villafuerte, Philip N Ainslie, and Damian Miles Bailey

Thank you for submitting your manuscript to The Journal of Physiology. It has been assessed by a Reviewing Editor and by 2 expert Referees and I am pleased to tell you that it is considered to be acceptable for publication following satisfactory revision.

The reports are copied at the end of this email. Please address all of the points and incorporate all requested revisions, or explain in your Response to Referees why a change has not been made.

NEW POLICY: In order to improve the transparency of its peer review process The Journal of Physiology publishes online as supporting information the peer review history of all articles accepted for publication. Readers will have access to decision letters, including all Editors' comments and referee reports, for each version of the manuscript and any author responses to peer review comments. Referees can decide whether or not they wish to be named on the peer review history document.

Authors are asked to use The Journal's premium BioRender (<https://biorender.com/>) account to create/redraw their Abstract Figures. Information on how to access The Journal's premium BioRender account is here: <https://physoc.onlinelibrary.wiley.com/journal/14697793/biorender-access> and authors are expected to use this service. This will enable Authors to download high-resolution versions of their figures. The link provided should only be used for the purposes of this submission. Authors will be charged for figures created on this premium BioRender account if they are not related to this manuscript submission.

I hope you will find the comments helpful and have no difficulty returning your revisions within 4 weeks.

Your revised manuscript should be submitted online using the links in Author Tasks: Link Not Available.

Any image files uploaded with the previous version are retained on the system. Please ensure you replace or remove all files that have been revised.

REVISION CHECKLIST:

- Article file, including any tables and figure legends, must be in an editable format (eg Word)
- Abstract figure file (see above)
- Statistical Summary Document
- Upload each figure as a separate high quality file
- Upload a full Response to Referees, including a response to any Senior and Reviewing Editor Comments;
- Upload a copy of the manuscript with the changes highlighted.

- A potential 'Cover Art' file for consideration as the Issue's cover image;
- Appropriate Supporting Information (Video, audio or data set https://jp.msubmit.net/cgi-bin/main.plex?form_type=display_requirements#supp).

To create your 'Response to Referees' copy all the reports, including any comments from the Senior and Reviewing Editors, into a Word, or similar, file and respond to each point in colour or CAPITALS and upload this when you submit your revision.

I look forward to receiving your revised submission.

If you have any queries please reply to this email and staff will be happy to assist.

Yours sincerely,

Professor Laura Bennet
Senior Editor
The Journal of Physiology
<https://jp.msubmit.net>
<http://jp.physoc.org>
The Physiological Society
Hodgkin Huxley House
30 Farringdon Lane
London, EC1R 3AW
UK
<http://www.physoc.org>
<http://journals.physoc.org>

EDITOR COMMENTS

Reviewing Editor:

Thank you for providing a thorough response to referees' comments and for providing a revised manuscript. Referee 1 raises some additional important points that should be addressed. Please respond to these final points and provide a revised manuscript which I look forward to receiving.

Additionally, regarding the statement on the Declaration of Helsinki. Please indicate if the study was a registered trial, or alternatively include the statement: "...Declaration of Helsinki, except for registration in a database".

REFEREE COMMENTS

Referee #1:

Stacey et al. have made substantial changes to this revised manuscript, including a more thorough rationale for the study, new and updated data, revised interpretation of results, and a limitations section in the discussion.

I have a few remaining comments for the authors to consider:

- What is the justification for continuing to include the two participants with cognitive function scores below the threshold of 26? Could continued inclusion of the data from these two participants unnecessarily increase the variability of the reported responses?
- While a justification for not including female participants is provided, is it known if variations in testosterone (diurnal variation or inter-individual variation) also have an effect on redox regulation and cerebrovascular function? If this is not known, this should also be included as a discussion point, to balance the rationale for not including female participants.
- The authors have added the term "cerebral perfusion" throughout the manuscript. Technically, however, perfusion is not being measured (i.e., blood volume per unit volume of tissue over time). Rather, global cerebral blood flow is estimated from the measurements being made. I recommend revising this terminology throughout the manuscript.

- Regarding the simultaneous increases in both arterial pressure and CO₂ during the CO₂ reactivity test, how is it possible to differentiate between the impact of increasing CO₂ and increasing arterial pressure on cerebral blood flow when both are increasing simultaneously? While arterial pressure increases by a similar magnitude across all conditions and groups, that does not necessarily mean that it has an "equal" contribution to the subsequent increase in cerebral blood flow, when CO₂ is also increasing.

Referee #2:

The authors have sufficiently addressed all the issues I have raised.

END OF COMMENTS

1st Confidential Review

05-Oct-2022

Response to Referees

We extend our sincere thanks to the Reviewing Editor and both referees for taking the time to (re)review our manuscript. We appreciate their recommendations/comments and have provided a brief response to each of the queries including modifications to the revised text (R2) highlighted in red.

Reviewing Editor: Thank you for providing a thorough response to referees' comments and for providing a revised manuscript. Referee 1 raises some additional important points that should be addressed. Please respond to these final points and provide a revised manuscript which I look forward to receiving. Additionally, regarding the statement on the Declaration of Helsinki. Please indicate if the study was a registered trial, or alternatively include the statement: "...Declaration of Helsinki, except for registration in a database".

Our pleasure and thank you for your general comments. The manuscript has since been amended in line with the additional points raised by the referees (please see responses below). In terms of your specific point, to confirm, this study was not a registered trial, as clarified under Ethical Approval: "Participants provided written informed consent and all procedures conformed to the Declaration of Helsinki, except for registration in a database".

REFeree COMMENTS

Referee #1: Stacey et al. have made substantial changes to this revised manuscript, including a more thorough rationale for the study, new and updated data, revised interpretation of results, and a limitations section in the discussion. I have a few remaining comments for the authors to consider.

Thank you for providing these additional comments, we greatly appreciate your time and efforts reviewing our manuscript. We have provided a brief response to each of the queries (below) including modifications to the revised text (R2) highlighted in red.

What is the justification for continuing to include the two participants with cognitive function scores below the threshold of 26? Could continued inclusion of the data from these two participants unnecessarily increase the variability of the reported responses?

Thank you for raising this point. Despite these two participants (1 lowlander and 1 highlander) achieving a MoCA score that is 1-2 points below the threshold for mild cognitive impairment, we believe that they are representative of a normative population and therefore their continued inclusion is justified in the present study. This decision is supported by Rossetti *et al.* (2011) who documented MoCA scores of 22.80 (\pm 3.38) points (mean \pm SD) in similarly aged adults (<35 years) with < 12 years of education and 25.93 (\pm 2.48 points) with > 12 years of education. Our MoCA data, including the (biological) variation around the mean, would thus be considered to be within the normative range for this population. Further, it should be noted that these participants were all considered apparently healthy since they did not meet any of our exclusion criteria including overt cardiopulmonary disease (hypertension, diabetes, hypertriglyceridemia, chronic obstructive pulmonary disorder), significant developmental delay or learning difficulties, diagnosis of any central neurological disease (aneurysm, stroke, transient ischemic attack, epilepsy, multiple sclerosis) or psychiatric disorders including any history of traumatic brain injury: it is therefore unlikely that the inclusion of these participants have influenced the variability of the reported primary outcomes. Additionally, given that it was one participant from each group, this balance likely negates any impact on the comparisons made between groups on

the series of measurements obtained. For further clarity, we have included the following paragraph in the limitations section “Despite rigorous medical screening and attainment of all inclusion criteria, two participants (one lowlander and one highlander) were diagnosed with borderline mild cognitive impairment (MCI) according to established criteria (Nasreddine *et al.*, 2005), reflected by MoCA scores of 24-25 points. Despite this, we specifically chose to not exclude these participants given that they were apparently healthy albeit with (slightly lower) scores that fell within the boundaries of normal (biological) variation expected from this population for their given age and level of educational (Rossetti *et al.*, 2011)”.

While a justification for not including female participants is provided, is it known if variations in testosterone (diurnal variation or inter-individual variation) also have an effect on redox regulation and cerebrovascular function? If this is not known, this should also be included as a discussion point, to balance the rationale for not including female participants.

Thank you for this comment. Upon reflection, this is an important point although we did not measure basal circulating testosterone levels. This limitation has since been acknowledged in the limitations section:

Equally, emerging evidence suggests that an inverse relationship may exist between circulating testosterone and oxidative-inflammatory stress (Maggio *et al.*, 2006; Rovira-Llopis *et al.*, 2017), notwithstanding the ability of testosterone to effect smooth muscle vasodilation directly (Webb *et al.*, 1999) and upregulate aquaporin-4 expression on astrocytic endfeet (Gu *et al.*, 2003) that could conceivably impact cerebrovascular function. However, it remains unclear to what extent basal differences in testosterone could have influenced the findings of the present study given the controversy reported in the literature in both lowlanders (Friedl *et al.*, 1988; Vasankari *et al.*, 1993) and native highlanders (Garmendia *et al.*, 1982; Gonzales *et al.*, 2009) exposed to high-altitude.

The authors have added the term "cerebral perfusion" throughout the manuscript. Technically, however, perfusion is not being measured (i.e., blood volume per unit volume of tissue over time). Rather, global cerebral blood flow is estimated from the measurements being made. I recommend revising this terminology throughout the manuscript.

Thank you for this recommendation. The term “cerebral perfusion” has since been amended to “cerebral blood flow [CBF]” throughout.

Regarding the simultaneous increases in both arterial pressure and CO₂ during the CO₂ reactivity test, how is it possible to differentiate between the impact of increasing CO₂ and increasing arterial pressure on cerebral blood flow when both are increasing simultaneously? While arterial pressure increases by a similar magnitude across all conditions and groups, that does not necessarily mean that it has an "equal" contribution to the subsequent increase in cerebral blood flow, when CO₂ is also increasing.

Thank you for this comment although it was not our original intent to differentiate between these competing stimuli. To the contrary, our primary goal was to unabashedly focus on the ‘global’ response in terms of documenting basal reactivity *per se*. However, to address your helpful point, we have since calculated the change in cerebrovascular conductance index (CVCi, calculated as CBF or CBV/MAP) during the hypercapnic challenge across all groups and included these data for your perusal (Table below): hence CBF has been ‘normalised’ to both (hypercapnia-induced elevations in) PET_{CO₂} and MAP. Given the small

sample sizes, were not in a position to run a MANCOVA to address the independent contributions of PET_{CO_2} vs. MAP to the global CVR_{CO_2} response (inadequate statistical power) and it is equally important to acknowledge that these simple calculations assume linearity, that we know to be incorrect (Ainslie & Duffin, 2009).

Table 1. MAP-normalised CVR_{CO_2} responses

	Lowlanders (n = 9)		Highlanders (n = 9)
	Lifelong-SL	Chronic-HA	Lifelong-HA
MCA CVCi CVR_{CO_2} ($cm \cdot s^{-1} \cdot mmHg^{-1} [MAP] \cdot mmHg^{-1} [PETCO_2]$)	3.51 ± 1.61	4.81 ± 5.79	3.05 ± 1.73
PCA CVCi CVR_{CO_2} ($cm \cdot s^{-1} \cdot mmHg^{-1} [MAP] \cdot mmHg^{-1} [PETCO_2]$)	3.24 ± 2.23	3.28 ± 3.12	2.93 ± 2.03
ICA CVCi CVR_{CO_2} ($mL \cdot min^{-1} \cdot mmHg^{-1} [MAP] \cdot mmHg^{-1} [PETCO_2]$)	3.89 ± 2.03	4.29 ± 3.34	6.40 ± 2.57*
VA CVCi CVR_{CO_2} ($mL \cdot min^{-1} \cdot mmHg^{-1} [MAP] \cdot mmHg^{-1} [PETCO_2]$)	6.84 ± 4.97	7.64 ± 6.00	5.15 ± 2.58

*different compared to Lowlanders at Lifelong Sea-Level (Lifelong-SL)

These data suggest that altitude acclimatisation failed to alter regional CVCi in the lowlanders (all comparisons, $P > 0.05$). In contrast, ICA CVCi was selectively elevated in the highlanders ($P = 0.046$, $d = 1.087$) implying a more 'reactive' pressure-induced elevation in (hypercapnia-induced) cerebral perfusion that may potentially confer a selective advantage to lifelong hypoxia, notwithstanding the aforementioned interpretive assumptions/limitations. A condensed appraisal of this additional information has been included in the Results and Discussion sections of the revised manuscript.

Referee #2: The authors have sufficiently addressed all the issues I have raised.

We thank you for your time and efforts in reviewing this manuscript.

Supporting References

- Ainslie PN & Duffin J. (2009). Integration of cerebrovascular CO₂ reactivity and chemoreflex control of breathing: mechanisms of regulation, measurement, and interpretation. *Am J Physiol Regul Integr Comp Physiol* **296**, R1473-1495.
- Friedl KE, Plymate SR, Bernhard WN & Mohr LC. (1988). Elevation of plasma estradiol in healthy men during a mountaineering expedition. *Horm Metab Res* **20**, 239-242.
- Garmendia F, Valdivia H, Castillo O, Ugarte N & Garmendia A. (1982). Hypothalamo-Hypophyso-Gonadal Response to Clomiphene Citrate at Median High-Altitude. *Hormone and Metabolic Research* **14**, 679-680.
- Gonzales GF, Gasco M, Tapia V & Gonzales-Castaneda C. (2009). High serum testosterone levels are associated with excessive erythrocytosis of chronic mountain sickness in men. *Am J Physiol Endocrinol Metab* **296**, E1319-1325.
- Gu F, Hata R, Toku K, Yang L, Ma YJ, Maeda N, Sakanaka M & Tanaka J. (2003). Testosterone up-regulates aquaporin-4 expression in cultured astrocytes. *J Neurosci Res* **72**, 709-715.
- Maggio M, Basaria S, Ble A, Lauretani F, Bandinelli S, Ceda GP, Valenti G, Ling SM & Ferrucci L. (2006). Correlation between testosterone and the inflammatory marker soluble interleukin-6 receptor in older men. *J Clin Endocr Metab* **91**, 345-347.
- Nasreddine ZS, Phillips NA, Bedirian V, Charbonneau S, Whitehead V, Collin I, Cummings JL & Chertkow H. (2005). The Montreal Cognitive Assessment, MoCA: a brief screening tool for mild cognitive impairment. *Journal of the American Geriatric Society* **53**, 695-699.
- Rossetti HC, Lacritz LH, Cullum CM & Weiner MF. (2011). Normative data for the Montreal Cognitive Assessment (MoCA) in a population-based sample. *Neurology* **77**, 1272-1275.
- Rovira-Llopis S, Banuls C, de Maranon AM, Diaz-Morales N, Jover A, Garzon S, Rocha M, Victor VM & Hernandez-Mijares A. (2017). Low testosterone levels are related to oxidative stress, mitochondrial dysfunction and altered subclinical atherosclerotic markers in type 2 diabetic male patients. *Free Radic Biol Med* **108**, 155-162.
- Vasankari TJ, Rusko H, Kujala UM & Huhtaniemi IT. (1993). The Effect of Ski Training at Altitude and Racing on Pituitary, Adrenal and Testicular Function in Men. *Eur J Appl Physiol* **66**, 221-225.

Webb CM, McNeill JG, Hayward CS, de Zeigler D & Collins P. (1999). Effects of testosterone on coronary vasomotor regulation in men with coronary heart disease. *Circulation* **100**, 1690-1696.

Dear Professor Bailey,

Re: JP-RP-2022-283362R2 "Lifelong exposure to hypoxia in humans is associated with improved redox homeostasis and structural-functional adaptations of the neurovascular unit" by Benjamin S Stacey, Ryan L Hoiland, Hannah G Caldwell, Connor A Howe, Tyler D Vermeulen, Michael M Tymko, Gustavo A Vizcardo-Galindo, Daniela Bermudez, Romulo Figueroa-Mujica, Chris Gasho, Eduoard Tuailon, Christophe Hirtz, Sylvain Lehmann, Nicola Marchi, Hayato Tsukamoto, Francisco C Villafuerte, Philip N Ainslie, and Damian Miles Bailey

We are pleased to tell you that your paper has been accepted for publication in The Journal of Physiology.

Authors should note that it is too late at this point to offer corrections prior to proofing. The accepted version will be published online, ahead of the copy edited and typeset version being made available. Major corrections at proof stage, such as changes to figures, will be referred to the Editors for approval before they can be incorporated. Only minor changes, such as to style and consistency, should be made at proof stage. Changes that need to be made after proof stage will usually require a formal correction notice.

Yours sincerely,

Professor Laura Bennet
Senior Editor
The Journal of Physiology
<https://jp.msubmit.net>
<http://jp.physoc.org>
The Physiological Society
Hodgkin Huxley House
30 Farringdon Lane
London, EC1R 3AW
UK
<http://www.physoc.org>
<http://journals.physoc.org>

P.S. - You can help your research get the attention it deserves! Check out Wiley's free Promotion Guide for best-practice recommendations for promoting your work at www.wileyauthors.com/eeo/guide. You can learn more about Wiley Editing Services which offers professional video, design, and writing services to create shareable video abstracts, infographics, conference posters, lay summaries, and research news stories for your research at www.wileyauthors.com/eeo/promotion.

IMPORTANT NOTICE ABOUT OPEN ACCESS: To assist authors whose funding agencies mandate public access to published research findings sooner than 12 months after publication, The Journal of Physiology allows authors to pay an Open Access (OA) fee to have their papers made freely available immediately on publication.

You can check if your funder or institution has a Wiley Open Access Account here: <https://authorservices.wiley.com/author-resources/Journal-Authors/licensing-and-open-access/open-access/author-compliance-tool.html>.

EDITOR COMMENTS

Reviewing Editor:

Thank you for making the final revisions to the manuscript. Congratulations on the completion of a very interesting study revealing new facets to high altitude adaptation, with evidence of greater neuroprotection to high altitude stress in Andeans native to high altitude, compared to 'sea-level' natives sojourning to high altitude.

REFEREE COMMENTS

Referee #1:

Thank you to the authors for your comprehensive responses to my remaining concerns. Congratulations on this very interesting manuscript!

2nd Confidential Review

28-Nov-2022